# Multi-batch single-cell comparative atlas construction by deep learning disentanglement

Allen W. Lynch[1,2], Myles Brown [3,4] & Clifford A. Meyer [2,5] ✉

Cell state atlases constructed through single-cell RNA-seq and ATAC-seq analysis are powerful tools for analyzing the effects of genetic and drug treatment-induced perturbations on complex cell systems. Comparative analysis of such atlases can yield new insights into cell state and trajectory alterations. Perturbation experiments often require that single-cell assays be carried out in multiple batches, which can introduce technical distortions that confound the comparison of biological quantities between different batches. Here we propose CODAL, a variational autoencoder-based statistical model which uses a mutual information regularization technique to explicitly disentangle factors related to technical and biological effects. We demonstrate CODAL's capacity for batch-confounded cell type discovery when applied to simulated datasets and embryonic development atlases with gene knockouts. CODAL improves the representation of RNA-seq and ATAC-seq modalities, yields interpretable modules of biological variation, and enables the generalization of other count-based generative models to multi-batched data.

Cell state atlases constructed with single-cell RNA-seq, ATAC-seq and multimodal technologies reveal a multiplicity of stable states and interconnected differentiation trajectories[1-6]. Combined with perturbations, including gene knockouts[7-9], drug treatments[10,11] and mutations[12,13], atlases can bring to light deep insights into the roles of transcription factors and signaling pathways in developmental processes, cancer plasticity, and other cell state transitions[8,14,15]. Perturbations can cause alterations in the structure of cell state atlases, including changes in cell population distributions, depletions, or enrichments of certain cell states, and possibly the emergence of wholly new cell states and differentiation trajectories. Comparing changes in cell population structures with unperturbed biology enables the mechanistic dissection of perturbation effects. However, perturbation experiments often require single-cell assays to be carried out in multiple batches, which can introduce technical distortions to the data[16]. This is especially problematic when batches contain different cell state population distributions or capture new and different cell types, which leads to technical effect confounding of the underlying batch-specific biological variation. Because current batch correction methods perform poorly when presented with batch-confounded cell states, cells in perturbation atlases are not typically compared directly, but are analyzed via projections onto reference wild-type cell state maps[8,15]. This practice precludes the possibility of directly observing alternative cell states and differentiation trajectories related to perturbations. Thus, batch correction methods that can remove technical artifacts while preserving biological cell states, especially those which are batch-confounded, would greatly enhance the power of comparative single-cell analysis.

To mitigate confounding technical effects, several batch correction methods have been proposed[17-19] which fall broadly into the categories of generative and latent space merging models. Most generative models for batch effect removal are derivatives of factor analysis or the variational autoencoder (VAE) framework. This includes VAE-based scVI[20], which has been shown to be one of the most

[1]Department of Biomedical Informatics, Harvard Medical School, Boston, MA, USA. [2]Department of Data Science, Dana-Farber Cancer Institute, Boston, MA, USA. [3]Center for Functional Cancer Epigenetics, Dana-Farber Cancer Institute, Boston, MA, USA. [4]Department of Medical Oncology, Dana-Farber Cancer Institute, Brigham and Women's Hospital, and Harvard Medical School, Boston, MA, USA. [5]Department of Biostatistics, Harvard T.H. Chan School of Public Health, Boston, MA, USA. ✉e-mail: cliff_meyer@ds.dfci.harvard.edu

effective approaches in single-cell RNA-seq atlas benchmarking tests[21]. scVI parametrizes the distribution of observed counts using a deep neural network conditioned on the joint distribution of latent variables and cell batch labels but makes no attempt to identify or separate the biological from the technical components of those observations. Other generative model-based methods include the semi-supervised VAE scANVI[18] and factor analysis-based ZINB-WaVE[22]. In addition, methods that use the generative adversarial network and maximum mean discrepancy frameworks[23] have been proposed, but these models require that each batch has the same cell population distribution. This is both difficult to assess *apriori* and inappropriate to assume when integrating perturbation datasets as considered here.

Alternatively, most effective latent space merging models use mutual nearest neighbors between datasets to find shared cell states between batches. The MNNs are then used to calculate a nonlinear projection to reduce the distance between batches in some latent space. This class includes popular methods such as Seurat v3[24], FastMNN[17], and Scanorama[25]. Finally, the Harmony[19] method removes technical effects from data by using cross-dataset fuzzy clustering to iteratively merge clusters of cells predicted to be in similar states. Neither current generative models nor latent space merging methods admit a direct explanation of how technical distortions influence the data. Unregularized estimations of technical effects can lead to over-correction or the misidentification of biological signals as technical effects. This may be why these methods struggle to detect batch-confounded cell states and are insensitive to differences in cell population distributions across batches.

Another important aspect of comparative atlas analysis is the deconvolution of the effects of different perturbations. Typically, analysis of single-cell RNA-seq and ATAC-seq atlases involves the representation of high-dimensional data in terms of low-dimensional latent spaces. Topic models[26–28] and matrix factorization methods[29,30] infer interpretable latent space representations that correspond to modules of co-regulated genes or co-accessible peaks. These modules may be useful for capturing and explaining the influence of different perturbations on gene regulatory programs. When considering batched data, however, differences in technical factors induced by single-cell experimental protocols impose a layer of batch-dependent distortion over the modular biological effects. Naively applied, matrix factorization methods cannot distinguish between biological and technical sources of variation, which results in the discovery of modules that may be contaminated by technical effects.

To mitigate current challenges in perturbation atlas analysis, we propose a variational autoencoder[31]-based statistical model and novel parameter inference procedure which extends interpretable topic modeling to batched single-cell data. This method, called CODAL (COvariate Disentangling Augmented Loss) explicitly disentangles factors related to technical and biological effects, decomposes biological effects into interpretable modules, detects batch-confounded cell states, and represents cells in a batch-corrected, yet cell-type discriminative, latent space. Our approach can be applied to single-cell RNA-seq, ATAC-seq, and to each modality within true multimodal RNA-seq plus ATAC-seq data. We benchmark the method using rigorously defined standards[32] and demonstrate its capacity for batch-confounded cell type discovery when applied to simulated datasets and on embryonic development atlases with gene knockouts[8]. In the integrated regulatory analysis of true multimodal data[21], we show that CODAL batch correction improves the representation of RNA-seq and ATAC-seq modalities and enables the generalization of other count-based generative models to multi-batched data[33]. Overall, CODAL delivers robust technical effect correction and representation for datasets with varying degrees of confounded cell population differences, dataset size, and technical or biological complexity (Supplementary Table 1).

The CODAL model architecture extends MIRA[28], our earlier method for variational topic modeling of single-cell RNA-seq and ATAC-seq data. CODAL includes additional modules to facilitate batch effect correction using our new objective function. Furthermore, we designed new highly scalable and parallelizable automated hyper-parameter selection and model training procedures which tailors the CODAL method to the properties of the dataset at hand. For fixed model size, CODAL training time scales linearly with the number of cells modeled, while memory usage is kept constant using an efficient minibatch stochastic gradient descent algorithm which streams data from a fast-loading on-disk cache (Supplementary Fig. 1). CODAL is available as an open-source Python package at https://mira-multiome.readthedocs.io.

## Results

### Framework for disentangling biological and technical effects

In the analysis of a multi-batch single-cell RNA-seq or ATAC-seq experiment, we explicitly decompose the variation in observed read counts into biological and technical components (Fig. 1a). We use a variational autoencoder-based implementation of Latent Dirichlet Allocation[34] to further factorize biological variation into latent variables, or "topics", $Z$, and matrix of linear feature associations, $\beta$. The topic compositions form a low-dimensional latent space that represents the cell states observable in the given population of cells. The linear association matrix constitutes modules of covarying biological quantities, gene expression or chromatin accessibility, that are evident in the data.

Technical effects are well known to confound the interpretation and comparison of single-cell datasets[35]. Although various types of technical distortions in RNA-seq and ATAC-seq have been described[36,37], each step in long single-cell protocols can contribute to such biases. The overall effect of all technical artifacts has therefore not been systematically characterized. Nevertheless, cells from the same batch, sharing protocol conditions and reagents, are subject to more similar technical effects than cells from different batches. Therefore, we assume that variance in technical effects is driven primarily by hidden batch-specific factors that systematically alter the read counts observed in each batch of cells. Although we do not precisely know the identity or effects of the technical factors, the cells' batch of origin can be used as a proxy variable to indicate which cells were subjected to a protocol using common reagents and under similar conditions.

We also observe that some technical effects appear to depend on both batch and cell state. In other words, some cell types exhibit different degrees of technical effect, even within the same batch. This can arise from cell state-dependent differences in cell size, cytoplasmic or nuclear chemistry, or cell state abundance within a population. Controlling for these state-dependent confounders would render cell states independent of technical effects. Typically, however, these technical factors are unknown and exert their influence on observed counts as contextual interactions between cell state and batch. Taken together, we estimate the distribution of technical effects in each gene in each cell as a function of the cell state random variable and the batch covariate proxies (Fig. 1b). This function is implemented as a neural network because the functional form of technical effects is unknown.

### Mutual information-based disentanglement of technical and biological phenomena

We aim to learn the distribution of expression rates for cells in state $Z$ without confounding technical effects. However, from the dependency diagram of the model (Fig. 1c), due to the association between expression rates ($\lambda$) and technical effects ($t$) implied by their dependence on cell state ($Z$), we find $t$ and $\lambda$ are not independent. To estimate gene expression under these conditions, we therefore need to make an additional assumption about the relationship between technical effects and biological quantities. Here we propose to approximate the unconfounded distribution of biological quantities by penalizing the

dependence between biological quantities ($\lambda$) and technical effects ($t$), through regularization of the mutual information between their distributions. To implement this, we augment the evidence lower bound (ELBO) objective function[31] with a lower bound approximation of the

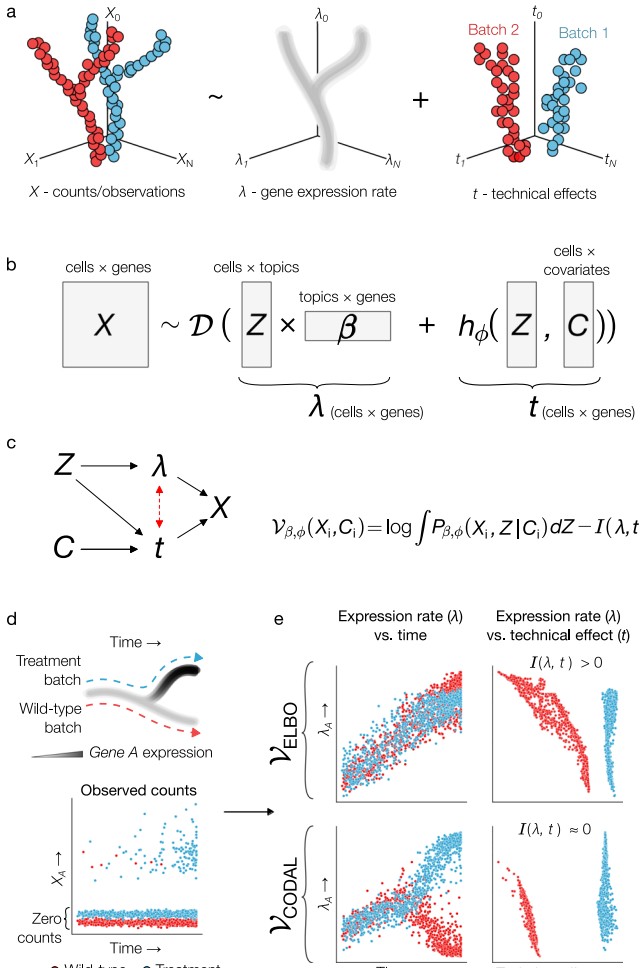

**Fig. 1 | CODAL disentanglement model overview and comparison to standard variational autoencoder. a** Decomposition of observed gene expression counts into biological and technical components. Gene expression follows some low-dimensional manifold, while technical effects are of an unknown functional form. **b** CODAL model description. $\mathcal{D}$ is the negative binomial distribution, $\lambda$ is the biological quantity (gene expression or chromatin accessibility), which is the product of cell latent topics $Z$ and linear gene associations $\beta$. The technical effect vector $t$ is given by a neural network $h$ with weights $\phi$, dependent on $Z$ and the cell covariates $C$ (batch of origin, quality control metrics, etc.). Counts are drawn from a distribution parameterized by the sum of biological quantity and technical effects. **c** (left) CODAL model structure. Dependencies between random variables are indicated by solid arrows. The red dash arrow indicates an association between $\lambda$ and $t$ that is implied by the dependence of $t$ on $Z$, confounding the direct estimation of the biological quantities. (right) The CODAL objective function maximizes the marginal log-likelihood of the data minus the mutual information ($I$) between biological quantities $\lambda$ and technical effects $t$. **d** Simulated bifurcating differentiation system with batch-confounded cell types. The treatment batch (blue) has elevated expression of gene $A$ relative to the wild-type batch (red) after a bifurcation in state. **e** Expression rates for gene A in simulated system estimated using model trained with ELBO versus CODAL objective. (top) The standard VAE objective (ELBO, marginal likelihood maximization only) yields poor estimates of gene A's expression level. Changes in expression are entangled with changes in technical effects. (bottom) Mutual information regularization disentangles the expression rate of gene $A$ from the technical effects, yielding expression estimates which match the data generation procedure. Using the CODAL objective, the mutual information between estimated gene expression and technical effects is minimal. Source data are provided as a Source Data file.

mutual information[38–40]. The result is a novel objective function we call the COvariate Disentangling Augmented Loss, or CODAL, which is a differentiable approximation of the sum of the mutual information and the marginal likelihood of the data. Optimizing with this objective yields a generative distribution that explicitly estimates unconfounded biological quantities in cells across batches and explains the influence of technical effects on observed counts in single-cell genomics experiments.

Penalizing mutual information between biological quantities and technical effects encourages the model to learn a function for technical effects which is largely independent of the cell state random variable $Z$, but still allows for modeling of those technical effects which do appear state-dependent. The CODAL objective, therefore, produces a generative explanation of the data which is a compromise between an idealized representation of technical effects which are assumed to be completely independent of the biological variation, and current methods where technical effect estimates are unconstrained and vary freely with cell state. When confronted with batch-confounded cell types (Fig. 1d, e), the practical implications of mutual information regularization are apparent: the CODAL objective enforces a distribution for technical effects that is not highly dependent on cell state, effectively disentangling the distributions of biological and technical effects, while marginal likelihood maximization (implemented using the "vanilla" ELBO objective) finds a complex technical effect function which confounds the coherent estimation of biological quantities.

To assess the effect of mutual information regularization on gene expression estimates in a batched scRNA-seq dataset, we compared a model with parameters estimated using the standard ELBO objective maximization to one estimated using the CODAL objective. The batched dataset, created for the 2021 NEURIPS Multimodal Single-Cell Data Integration challenge[33], was generated by distributing bone marrow samples from multiple donors to multiple laboratories for single-cell RNA-seq and ATAC-seq analysis. The resulting dataset has a hierarchical batch structure, where multiple donors were assayed in different batches at more than one site. Without mutual information regularization, we find that technical effects and expression changes are frequently correlated or anti-correlated. With increasing observed counts, the model confounds batch effects and biological changes (Fig. 2a). For the *Ccl5* gene, a marker for NK and CD8+ T cells[33,41], the unregularized model erroneously predicted expression to be highest in erythroblast cells. The monocyte marker *Tcf7l2*[33,42] likewise shows a pattern of high expression in some cell types that is not supported by the observed counts. Importantly, the solutions produced from optimizing the ELBO objective yield incoherent and entangled explanations for the observed data.

In contrast, mutual information regularization yields uncorrelated and disentangled expression rates and technical effects for each marker gene (Fig. 2b), where the known associated cell types are predicted as having the highest expression. Notably, the marginal distribution of expression rates within each cell type is similar across batches, so the cell types are arrayed vertically according to the relative expression levels of genes within these cell types. Batches, meanwhile, are arrayed horizontally according to the basal levels of counts observed within them. The CODAL objective therefore disentangles biological and technical contributions to observed counts and represents those contributions as independent factors.

## CODAL latent space demonstration of strong performance on cell-type discrimination and batch correction benchmarks

Applied to the 2021 NEURIPS dataset, CODAL successfully merges batches, finds shared cell types, and distributes those cell types along known paths of hematopoietic differentiation (Fig. 3a, b). Evaluating the quality of the latent space using silhouette width with expert-annotated cell types, we found that cells with the same label tend to be closer together in the CODAL-derived latent space than in the space

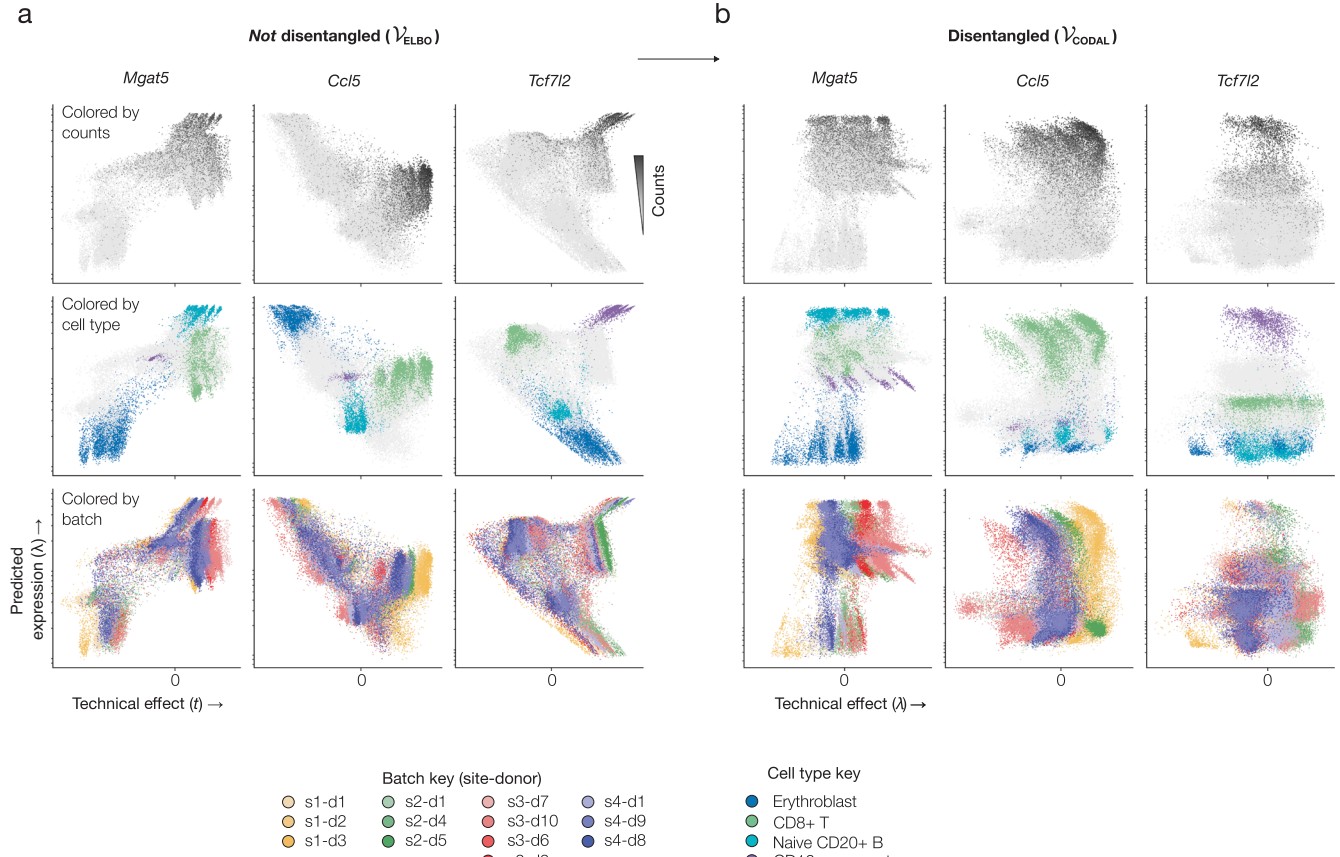

**Fig. 2 | Mutual information regularization disentangles the influence of biological and technical effects. a** Predicted gene expression rates versus predicted technical effects for naïve B-cell, NK/CD8+ T-cell, and monocyte marker genes estimated using a topic model trained with the marginal likelihood maximization objective ($\mathcal{V}_{ELBO}$). Colored by observed expression counts, expert-annotated cell types, and batches. The batches are labeled according to single-cell assay site (s1-s4) and sample donor (d1-d10). **b** Predicted expression rates versus technical effects for the same genes, estimated using a topic model trained with mutual information regularization ($\mathcal{V}_{CODAL}$). Source data are provided as a Source Data file.

derived by scVI[20], a more traditional VAE model which uses likelihood maximization with unconstrained technical effect modeling (Fig. 3c). Next, we benchmarked CODAL against popular batch correction methods shown to be effective in an extensive benchmarking study[21] (Fig. 3d, Supplementary Figs. 2 and 3). Notably, CODAL demonstrated both batch mixing and cell type colocalization comparable to scANVI, which previously demonstrated best-in-class atlas-level integration[21] and was fully supervised on cell type label for this test. Through regularization of the technical effect function, CODAL yields more discriminative biological latent space descriptors than VAE and geometry-based models without loss of capacity for technical effect correction.

In addition to yielding a well-separated latent space, CODAL's latent dimensions are designed to be interpretable, unlike those of deep latent variable models. CODAL's use of a sparsity-inducing hierarchical Dirichlet prior results in latent dimensions that coherently convey changes in cell-type identity and potentially deconvolve contributions of gene regulatory programs (Fig. 3e, Supplementary Fig. 4). Each latent variable, or topic, is linearly associated with changes in gene expression through the $\beta$ matrix. These sets of associations capture some covarying element, or module, of gene expression. Topic 14, for instance, precisely described the CD16+ monocyte identity program (Fig. 3f, g), and the captured associations were well correlated with the log-fold change of those genes' expression in CD16+ monocytes relative to the rest[43]. While differential gene expression and log-fold change are usually defined by investigator-driven clustering after latent space construction, CODAL topics and their consequent relationships with gene expression changes are learned jointly and directly from the data.

Since the bone marrow dataset is multimodal, assaying gene expression and chromatin accessibility in the same single cells, we also carried out the benchmarking analysis on the ATAC-seq data (Supplementary Figs. 5 and 6a, b), finding CODAL to perform well by several metrics. In addition to categorical indicators of batch, CODAL allows for the inclusion of continuous proxies for technical effects (Supplementary Figs. 6c, d and 7). One possible proxy is the FRiP score (fraction of reads in peaks), a commonly used ATAC-seq quality control metric[44]. scATAC-seq data is typically analyzed by first calling "peaks", or frequently accessible loci, from the aggregate profile of reads sequenced across all cells. Individual cells are encoded as vectors of binary variables, indicating the presence or absence of fragments in peaks. The FRiP score is used to identify and remove cells for which observed reads are primarily from noisy background genomic regions and do not contribute to the aggregate read peaks. A peak set derived from the bulk signature of a batched sample will tend to be biased toward the most common cell types and the largest batches, which can influence the distribution of peaks observed in each cell in a jointly batch and cell-specific manner. The FRiP score is therefore an example of a cell state-dependent technical factor in single-cell ATAC-seq analysis. Including the FRiP score in the CODAL analysis, in addition to batch indicators, resulted in a slight improvement in performance (Supplementary Fig. 6a, b) and reduced separation of cell type subpopulations which were affected by large differences in FRiP (Supplementary Fig. 6c). This suggests that further improvements to technical effect disentanglement might be attainable through careful characterization of sources of technical bias in single-cell analysis protocols.

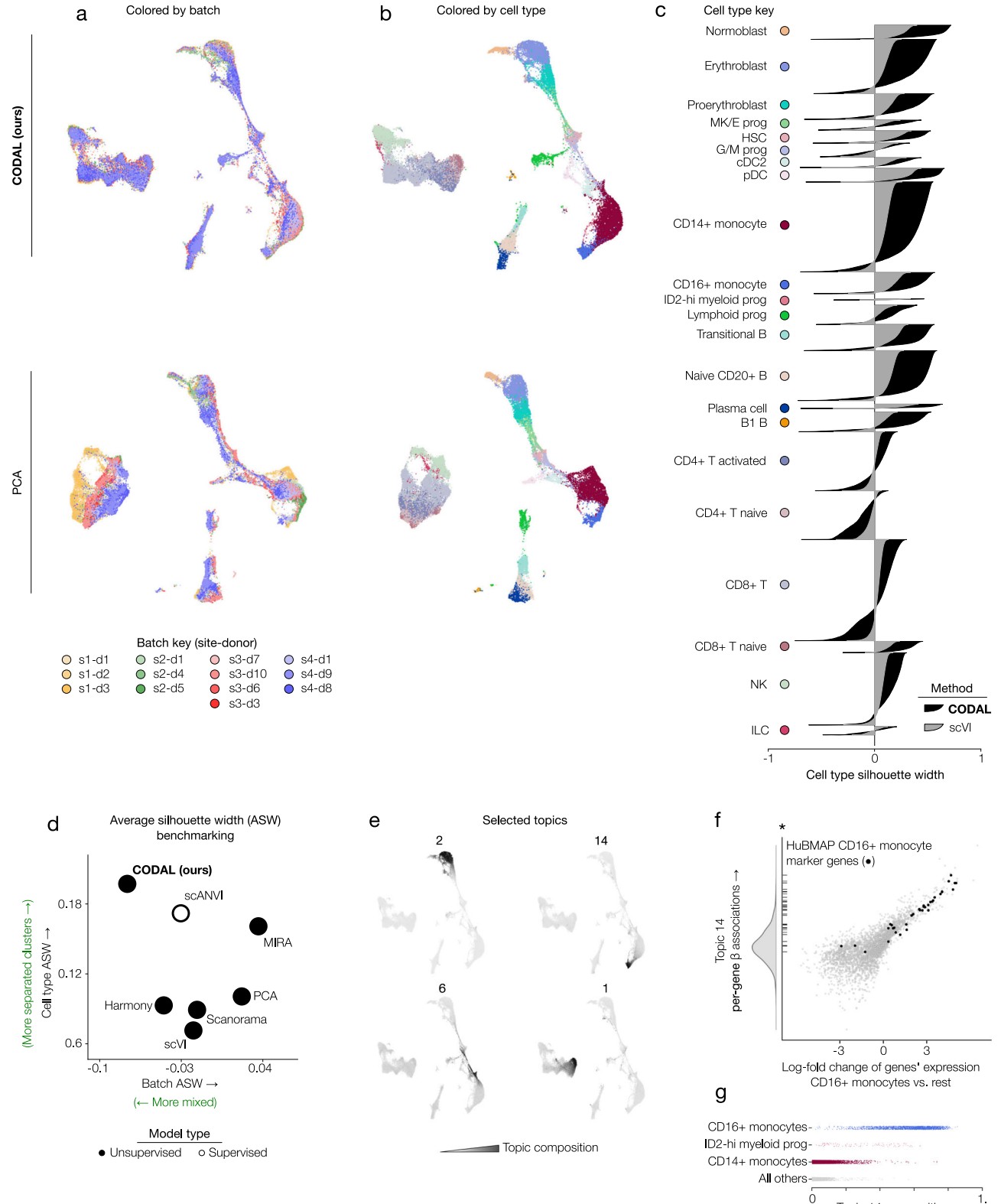

## CODAL-enhanced refinement of regulatory models connecting chromatin accessibility and gene expression

While CODAL's batch-corrected latent space estimation is useful for constructing and merging cell state atlases, its estimates of gene expression, chromatin accessibility and technical effects can be used to improve other types of single-cell analysis. Since mutual information regularization renders CODAL technical effect estimates effectively independent of the biological variation underlying changes in gene expression, we hypothesized these technical effect estimates are "transferable". This means CODAL-estimated technical effects can be fixed in a second generative model that incorporates explanatory features that were not included in the primary CODAL analysis. The resultant generative distribution will adjust for the confounding technical effects, and its parameters can be estimated using standard likelihood maximization. Therefore, CODAL technical effect estimates could be used to extend other generative models to batched data.

**Fig. 3 | Benchmarking demonstrates CODAL's strong performance relative to popular batch effect correction algorithms. a** UMAP representations of NEURIPS bone marrow gene expression data based on CODAL and PCA latent spaces, colored by batch. The batches are labeled according to single-cell assay site (s1-s4) and sample donor (d1-d10). Principal component analysis (PCA) does not correct for technical effects. **b** UMAPs colored according to expert-annotated cell types. **c** Cell type silhouette per cell comparing CODAL to scVI latent spaces. Higher cell silhouette widths indicate that cells are closer to other cells with the same cell type label. **d** Benchmarks of cell type and batch average silhouette widths (ASW) calculated from latent spaces using multiple methods. Increasing cell type ASWs correspond to greater similarities between cells with the same annotations. Decreasing batch ASWs correspond to greater mixing of cells from different

batches. scANVI is a semi-supervised method that makes use of cell-type label inputs. MIRA is a baseline topic modeling algorithm that does not correct for technical effects. **e** Selected topic composition shown on the CODAL latent space-based UMAP. **f** Topic 14 associations (from $\beta$ matrix) versus gene expression log-fold changes for CD16+ monocytes relative to other cell types in the dataset. Genes are colored by inclusion in the HuBMAP CD16+ monocyte ontology (black) versus not (gray). The marginal distribution of $\beta$ associations is shown as a density plot on the y-axis, with a rug plot marking the genes in the CD16+ ontology. Genes in the CD16+ ontology were significantly enriched in the top 200 topic 14-associated genes (p-value=$1.5 \times 10^{-19}$, one-sided Fisher's exact test). **g** Topic 14 composition across CD16+ monocytes, CD14+ monocytes, myeloid progenitors, and other cell types. Source data are provided as a Source Data file.

For example, the MIRA software for the integrative analysis of multimodal single-cell RNA-seq and ATAC-seq includes a method for relating gene expression changes with cis-regulatory chromatin accessibility[28]. This method models the cis-regulatory environment of a gene as a regulatory potential, in which the influence a chromatin accessible genomic interval has on a gene's expression decays exponentially with the genomic distance between the transcription start site of the gene and the region itself. For each gene, MIRA estimates the decay rates upstream and downstream of the gene along with the relative activities of the upstream, downstream and promoter regions (Fig. 4a). In the previous implementation of this method, the model parameters could be learned from only one batch of single-cell multimodal data as technical effect confounders from both the RNA and ATAC-seq batches would obscure the regulatory relationship or preclude its estimation. We augmented the regulatory potential model by adding CODAL technical effect vectors as a fixed aspect of its generative distribution, then applied the regulatory potential analysis to batches of 10x Genomics Multiome data comprising the bone marrow hematopoiesis dataset[33]. In principle, the regulatory ranges and activations estimated by the augmented model should be more accurate than the parameters predicted on non-batch-corrected data.

We trained uncorrected regulatory potential models on data from each batch alone and across all batches combined. To compare these with CODAL-corrected models trained across all batches, we computed the likelihoods of the models on held-out cells from each batch (Fig. 4b). Differences in technical effects confounded the application of models trained on one batch when tested on another. When trained across all batches without correction, the models suffered from high bias because variation in technical effects between samples was left unexplained. Finally, training across all batches with fixed CODAL technical effect correction vectors gave the most likely model across all batches and broadly across all genes (Fig. 4c), despite having identical trained parameters.

To determine if the improvements in likelihood translate to a better understanding of the biology of the system, we used a procedure called probabilistic in silico deletion to find JASPAR motifs[45] enriched within the influential chromatin regions prescribed by regulatory potential functions for 200 genes upregulated in proerythroblast cells (Fig. 4d, e). Regulatory potential models which better explain the true cis-regulation of these genes should contain more relevant motif hits. Without technical effect correction, the only significant motif enrichment (p-value < 0.05 (Bonferroni-corrected), n = 1600) was for TAL1-GATA1 complex motifs, which represent a key transcription factor complex promoting early erythropoiesis[46,47]. Regulatory potential models trained with CODAL technical effect correction resulted in 3 orders of magnitude increase in significance of TAL1-GATA1 enrichment in addition to significant enrichment for relevant GATA family and TAL1-TCF3 complex motifs[46].

These results suggest that the technical effects learned as part of CODAL modeling remove independent and confounding sources of technical variation in the data. Their incorporation into the generative distributions of other models enhances biological signal recovery in

multi-batch modeling, improving the performance and interpretability of these models even over single-batch applications.

## CODAL-enabled identification of batch-confounded cell types on Frankencell benchmarks

One of the most valuable uses of CODAL is its application to single-cell analyses in which cell types are confounded with batch. In studies of cell state plasticity and differentiation, for example, it is of interest to compare perturbations of biological systems, such as gene knockouts, with each other and with wild-type cells. Such projects are likely to be carried out in a series of batches, with discoveries made in early batches informing later experiments. We constructed gold-standard benchmarking datasets using Frankencell, a program that synthesizes cell differentiation trajectories by sampling reads and cells from real multimodal datasets (Fig. 5a). In these benchmarks we simulated datasets in which wild-type and perturbed cells were collected in different batches. The batch data was derived from real 10x Genomics Multiome batches of data, therefore the data sampled from these batches are affected by real technical effects. We modulated the difficulty of the tests by varying parameters defining the synthetic datasets, including the degree to which each of two terminally differentiated cell types are enriched or depleted from each batch, and mixing proportions determining the states of the synthesized cells. In this way, we varied the extent to which known biological variation was confounded by technical batch effects.

We used CODAL, PCA corrected with Harmony, and scVI, to construct latent space representations of the inferred biological states, then compared the resultant space to ground truth trajectories using established trajectory metrics implemented in the *dynverse* benchmarking package[48] (Fig. 5b, Supplementary Fig. 8a). The most difficult tests are "completely confounded", as the perturbation batch solely produces a new terminal cell type that is not present in the simulated wild-type batch. As expected given the limitations of likelihood-based models, scVI shows deteriorating performance in accordance with the level of batch-confounded biology in the test, while Harmony, which performs particularly well on smaller problems[21], has difficulties in the completely confounded samples (Fig. 5b, c). CODAL, meanwhile, is robust to all types of confounding while still merging shared cell types. On these benchmarks, CODAL never failed to solve the correct trajectory.

Next, we used a "completely confounded" Frankencell dataset to investigate how the mutual information regularization term of the CODAL objective function affects representation quality and repeatability in batch-confounded cell systems (Supplementary Fig. 8b). We varied the mutual information regularization strength by introducing a weight to that term of the objective function. For each regularizer strength, we modeled the same dataset ten times with different initial seeds, then assessed the quality of the reconstructed trajectory and the variance of the resulting technical effect estimates. We found modeling the dataset using CODAL's default regularization strength resulted in a 15-fold reduction in the variance of technical effect estimation versus unconstrained marginal likelihood maximization, while

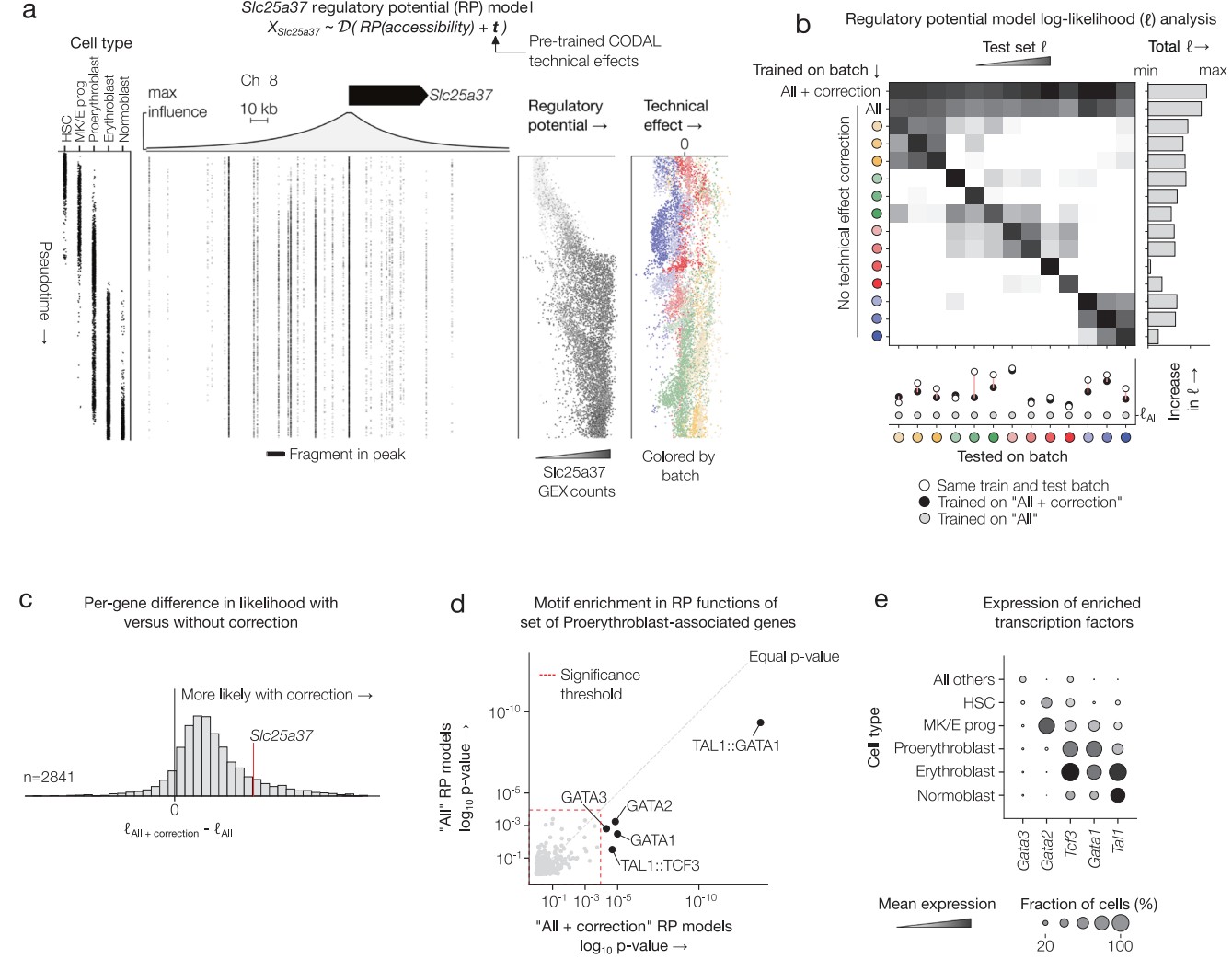

**Fig. 4 | Disentanglement improves the performance of other count-based single-cell models. a** Regulatory potential (RP) model relating *Slc25a37* proximal chromatin accessibility with gene expression. Model was trained while adjusting for both chromatin accessibility and gene expression technical effects using pre-trained technical effect vectors from the CODAL model. Fragment plot shows chromatin accessibility dynamics during differentiation from hematopoietic stem cells (HSCs) to normoblasts. Regulatory potential—the expression prediction given by the RP model—correlates with increasing accessibility of nearby loci and with observed expression counts. **b** Analysis of log-likelihood ($\ell$) of RP models trained with and without augmented technical effect correction. Each batch of the bone marrow dataset was split into training and testing datasets. For all highly variable genes, RP models were trained on the training sets of each batch individually, on all batches combined ("All"), or on all batches with technical effect correction ("All + correction"). The likelihood of each subsequent set of RP models was evaluated across each batch's test set. The cell colors in the heatmap are column-normalized. (right) The "Total $\ell$" gives the likelihood of that model across all held-out cells.

(bottom) For each batch, the increase in likelihood afforded by technical effect correction was compared against the likelihood of the uncorrected model trained only on that batch. Technical effects explain the difference in likelihood between models trained and tested within a batch versus across all batches. **c** Comparison of the likelihood of RP models trained with and without technical effect correction for each gene. **d** For the top 200 genes associated with the proerythroblast-specific topic 4, enrichment of transcription factor motifs within open chromatin regions predicted to be influential on expression of each of those genes by regulatory potential models. Enrichment was calculated using the probabilistic in silico-deletion test (one-sided). The enrichment *p*-value is shown for each transcription factor when tested against regulatory potential models trained on all batches with technical effect correction ("All + correction") versus without ("All"). Significantly enriched motifs ($\alpha = 0.05$, Bonferroni adjusted) are labeled. **e** Gene expression of each factor whose motif was significantly enriched in (**d**), for each cell type in the erythroid lineage. Source data are provided as a Source Data file.

producing the best-reconstructed trajectories (Supplementary Fig. 8c). Of the total variance describing biological and technical effects across ten CODAL models, only 0.14% was attributable to repeated technical effect estimation.

## Integrated analysis of wild-type and *Tal1* knockout embryonic developmental atlases

Pijuan-Sala et al.[8] constructed an extensive single-cell RNA-seq atlas of mouse embryonic development spanning stages E6.5 to E8.5. In addition to the wild-type mouse, this study included an analysis of the impact of knocking out the transcription factor *Tal1*, which is

necessary for embryonic erythropoiesis. The complete knockout of *Tal1* is lethal in the mouse embryo, so they created a chimeric mouse embryo combining wild-type and *Tal1* knockout cells (*Tal1*[-/-]). Wild-type and chimeric embryo data were generated in separate batches, which exhibit severe batch effects on uncorrected data, precluding meaningful direct comparisons between chimeric and wild-type manifolds (Supplementary Fig. 9a). They therefore present an analysis based on the projection of the chimeric cells onto the wild-type manifold. In the resulting projection, no erythroid cells were identified with a *Tal1*[-/-] genotype, as expected[49], and the mapping resulted in an overabundance of cells mapped as hemato-endothelial progenitors,

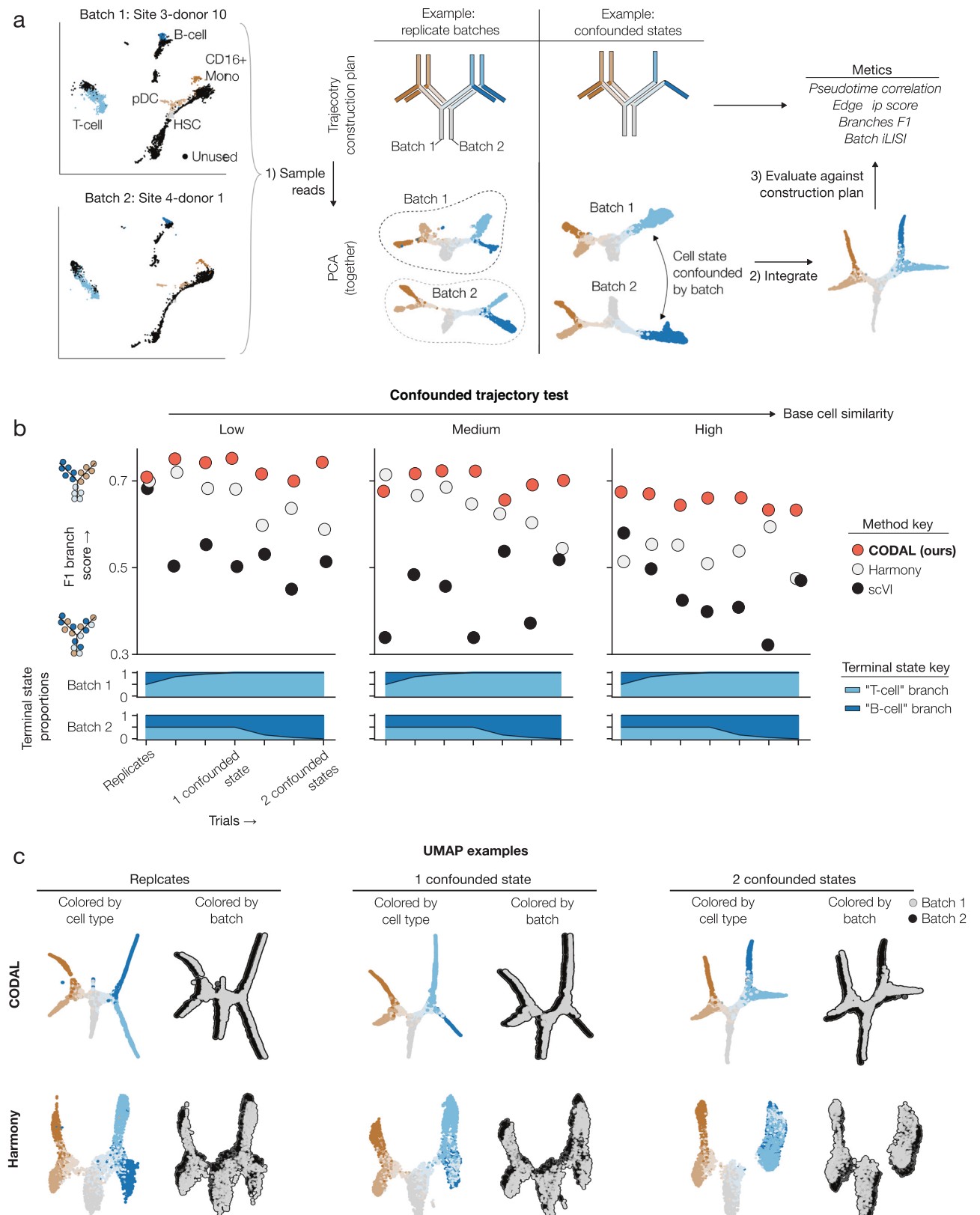

**Confounded trajectory test**

**UMAP examples**

suggesting the *Tal1⁻/⁻* cells were arrested in a state preceding *Tal1* activation. Their analysis of the projected hemato-endothelial progenitor cells revealed transcriptional differences from the wild-type cells with which they were co-embedded. Pijuan-Sala et al. had originally identified three *Tal1⁻/⁻*-specific hemato-endothelial progenitor subtypes which also expressed marker genes typically indicative of

major mesodermal populations. In particular, they found a mesenchyme-like population expressing marker *Tdo2*; an allantois-like population expressing *Pcolce*; and a population expressing cardiac-related genes *Nkx2-5*, *Mef2c*, and *Tnnt2*. The core assumption in the projection-based analysis is that the reference population comprises all cell types present in the projecting population. However, since the

**Fig. 5 | Mutual information-based disentanglement enables the discovery of batch-confounded cell types. a** "Frankencell" synthetic dataset generation and evaluation algorithm. Starting from either of two batches in the NEURIPS bone marrow dataset which exhibit different technical effects, reads from annotated cell clusters were mixed according to a construction plan to create a simulated differentiation trajectory interpolating between cell types. Datasets were composed of a trajectory constructed from both batches to represent known sources of biological and technical variation. By controlling terminal cell states present in each trajectory, we introduced batch-confounded cell types. We also varied the base cell similarity to measure method robustness. The trajectories were then integrated and evaluated against the construction plan using established trajectory comparison metrics. **b** Results from F1 branch score metric across all tests, colored by method. F1 branch score measures the similarity of predicted cell branch assignments to the construction plan. All branch assignments were calculated using MIRA pseudotime trajectory inference on the integrated trajectory. **c** Example UMAPs of CODAL and Harmony latent spaces, colored by true cell type and batch of origin. Cells from different batches were slightly offset for readability. Each example was taken from the "Medium" base cell similarity test. Source data are provided as a Source Data file.

*Tal1⁻/⁻* and wild-type hemato-endothelial progenitor cells showed transcriptional differences, there is an indication that cell states exist in the *Tal1⁻/⁻* cells that are rare or non-existent in the wild-type.

Reanalyzing this dataset with CODAL, we constructed a new latent space where the wild-type and chimeric *Tal1⁻/⁻* batches were modeled together (Fig. 6a). Whereas severe batch effects are present in the PCA-based analysis of the combined batches (Fig. 6b), the batches are well integrated in the CODAL analysis. Consistent with Pijuan-Sala et al., we found an abundance of *Tal1⁻/⁻* cells arrested in hemato-endothelial progenitor-*like* states (Fig. 6c). The CODAL latent space, however, revealed that while these cells were most similar to wild-type hemato-endothelial progenitor cells, they crucially occupied their own latent subspace and showed transcriptional signatures unseen in wild-type cells. After using the Leiden algorithm to cluster Tal1⁻/⁻ hemato-endothelial progenitors in the CODAL latent space, we identified five distinct subtypes (Fig. 6d) whose numbers were induced or promoted in the chimeric perturbation. We matched expression in these subtypes to wild-type mesodermal and endothelial subtypes through shared marker gene expression and topic composition (Fig. 6e, f, Supplementary Fig. 9b). Populations B, C and E expressed the aforementioned mesenchymal, cardiomyocyte, and allantois markers, respectively, corroborating the subtypes discovered in Pijuan-Sala et al. Furthermore, we found previously undescribed subtype A, with high expression of the endothelial marker *Emcn*, and ExE mesoderm-like subtype D, with high expression of homeobox transcription factors *Hoxb9*, *Hoxc8*, and *Cdx4*. Crucially, although each subcluster expressed distinct gene sets associated with different mesodermal cell populations, all subclusters expressed high levels of hemato-endothelial progenitor-specific markers *Kdr, Sox17*, and *Esam*[8]. Thus, the perturbation-induced subclusters exhibit combinations of cell identity programs that were not sampled in wild-type development.

The chimeric embryo system is a well-controlled system for studying perturbations because wild-type and perturbed cells can be compared within the same animal and batch. However, this approach cannot be applied to most types of perturbation experiments. As a further test of CODAL's ability to detect biologically confounded batches, we reanalyzed this dataset after excluding the control *Tal1⁺/⁺* wild-type cells from the chimeric mouse batch and removing samples that connected batches across time (Supplementary Fig. 10). Even in a dataset with contrived batch effect confounding, we find similar results as before (Fig. 6g, h).

Notably, we can reproduce the original ad hoc projection and differential expression analysis using only the CODAL model's inferred latent space and gene expression modules. In addition to CODAL latent spaces themselves being powerful representations of biological states, they are also useful for refining downstream analyses such as differential abundance testing and pseudotime trajectory analysis. Single-cell atlas construction using CODAL's batch correction method is therefore a powerful new strategy for making biological discoveries through the comparison of wild-type and perturbed systems.

## Discussion

We describe CODAL, a method for correcting technical batch effects in single-cell RNA-seq, ATAC-seq, and multimodal data while allowing for differences in biological states between batches. CODAL's variational autoencoder-based framework uses topic modeling to infer interpretable gene co-regulation and transcription factor binding modules in respective single-cell RNA-seq and ATAC-seq datasets. We introduce a novel mutual information-based regularization scheme that disentangles the contribution of biological modules and technical distortions to observed read counts and enables the estimation of gene expression and chromatin accessibility levels independent of a cell's batch of origin. Empirically, we demonstrate that CODAL produces latent representations of cell state which are both cell-type discriminative and batch-corrected, on par with current state-of-the-art supervised models.

CODAL is particularly well suited for identifying batch-confounded cell states that escape detection by current methodologies. In the comparison of cell state atlases in perturbed model systems, such as gene knockouts and drug treatments, this facilitates the discovery of different biological states and trajectories even when confounded by batch effects. In practice, CODAL enables the integrated analysis of experiments involving a series of batches, where observations from initial batches inform subsequent gene perturbations. This approach is a powerful and practical strategy for analyzing complex cell systems.

CODAL can robustly disentangle technical and biological effects in perturbation experiments when the effect of that perturbation varies according to cell state. To design experiments that satisfy this condition, a simple guideline is to include a control group of cells that do not experience or are known not to respond to the perturbation in both treatment and control batches. In differentiating systems, this may take the form of genetic perturbations which only affect cells in more differentiated states, as demonstrated in the Pijuan-Sala et al. and Frankencell analyses. Drug treatments may be integrated by multiplexing treatment samples with controls or again by measuring a group of cells that are known to be unaffected.

While this enables increased flexibility in experimental design, we caution that the existence of new batch-confounded or perturbation-induced cell states should be validated using orthogonal biological evidence to rule out the under-correction of technical effects. For example, bona fide cell states may have expression patterns that reflect known gene ontologies or demonstrate state-specific accessible chromatin which is enriched for transcription factor motifs. Although CODAL can overcome some of the challenges imposed by confounding batch effects, experimentalists nevertheless must practice good experimental design principles and include appropriate controls[50,51] as outlined above.

CODAL models variation in technical effects in a cell state-dependent manner. In addition to using batch identifiers as covariates for batch correction, CODAL can use other indicators of technical effects including single-cell RNA-seq and ATAC-seq quality control metrics, which could enable more accurate inference of chromatin accessibility or gene expression levels. In scATAC-seq data we discovered one such jointly cell-state and batch-dependent technical covariate, the FRiP score. When introduced into the CODAL model, the FRiP score improved technical effect removal in chromatin accessibility data. Further improvements in the single-cell analysis might be

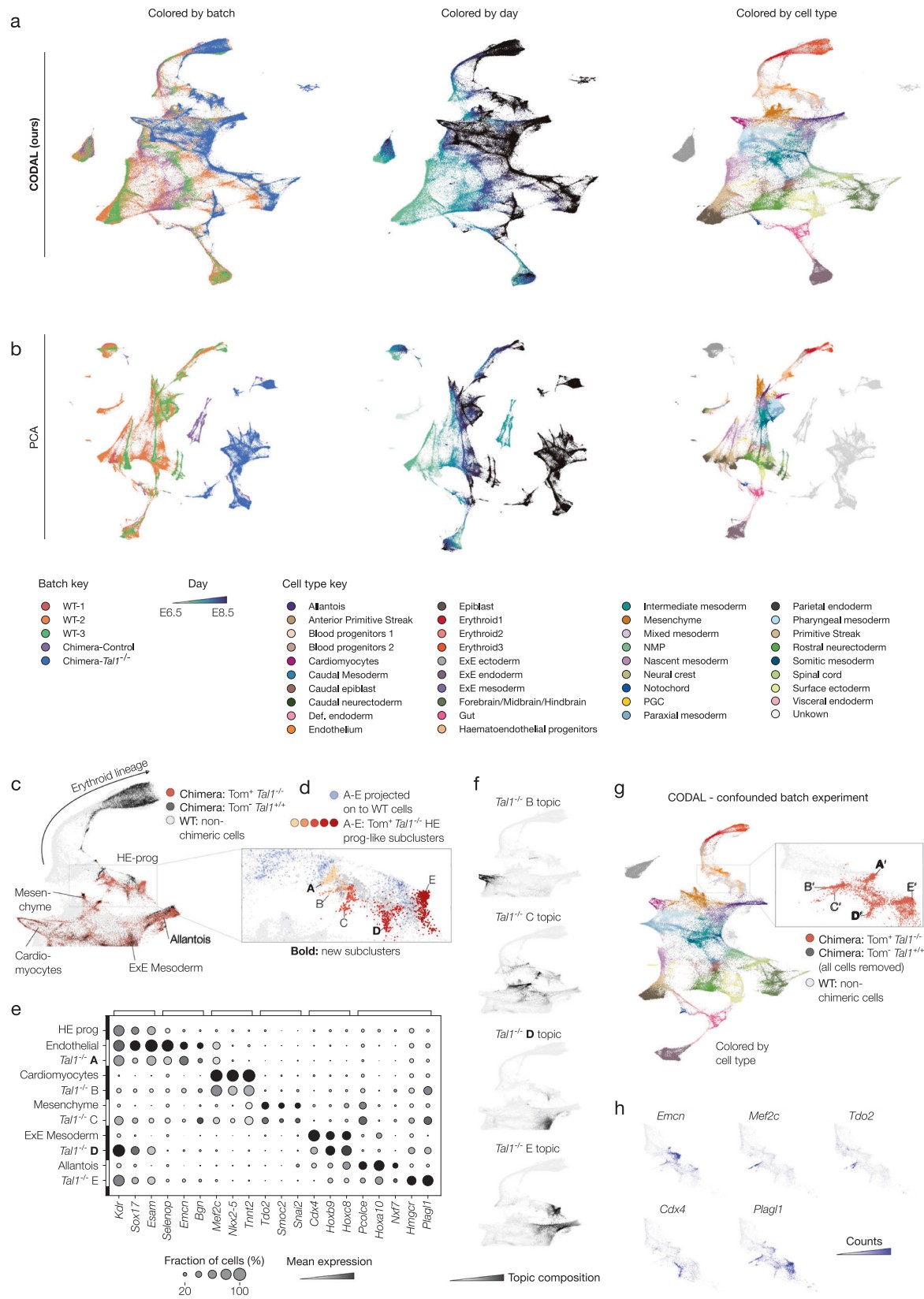

achieved through better characterization of the technical sources of bias in conjunction with the CODAL model.

Finally, CODAL explicitly estimates the contribution of technical effects in the generative distribution of read counts. Because the inferred technical effects are effectively independent of expression levels, one may use them to augment other generative models of read counts

with batch-corrective power. CODAL's correction of technical artifacts through multi-batch integration can therefore be used as a preprocessing step to improve the performance of other single-cell analysis methods. This could expand the application of generative models that are typically limited to single-batch analysis to include multi-batch datasets. Taken together, CODAL is a useful and much-needed tool for

**Fig. 6 | Re-analysis of mouse embryo differentiation using CODAL reveals new cell types in _Tal1_ knockout chimera. a** UMAP of CODAL latent space calculated from the mouse embryo differentiation dataset. From left to right, UMAP colored by cell batch of origin, day of differentiation at which cells were collected, and cell type annotations provided by the authors of the original study. Batches WT1-3 contained only WT cells. Chimera-Control embryos were injected with Tomato-expressing but otherwise unaltered embryonic stem cells (ESCs) early in differentiation. Chimera-_Tal1_$^{-/-}$ embryos were injected with Tomato-expressing _Tal1_$^{-/-}$ ESCs. **b** UMAP of PCA latent space. **c** CODAL UMAP colored by Tomato presence in Chimera-_Tal1_$^{-/-}$ batch. Tomato-expressing (Tom$^+$) cells have _Tal1_$^-$ genotype; Tom$^-$ cells have WT genotype. All other batches are shown in gray. **d** Zoom-in on hemato-endothelial progenitor (HE-prog) cell type population. New cell types present only in Tom$^+$ _Tal1_$^{-/-}$ perturbed population were subclustered using the Leiden algorithm and shown in shades of red. Previously uncharacterized subclusters are labeled in bold. From the original projection-based analysis, the closest WT cell to each _Tal1_$^{-/-}$ cell in clusters A-E are shown in blue. **e** Marker gene expression, column-normalized, comparing gene expression in WT populations with _Tal1_$^-$ HE-prog subclusters. **f** Selected topic compositions shown on UMAP. Shared topics indicate covarying gene expression between _Tal1_$^-$ subclusters and specific mesodermal cell types. **g** (left) UMAP of CODAL latent space calculated from embryo dataset with contrived batch structure maximizing confounded biology and technical variation. (right) Zoom-in on HE-prog cell type population, colored by Tomato expression in Chimera-_Tal1_$^{-/-}$ batch. All other batches are shown in gray. Tom$^-$ cells were removed to increase confounding effects. **h** Marker gene expression shown on HE-prog cell type subset of UMAP from (**g**). Source data are provided as a Source Data file.

the study of comparative single-cell atlases and for the mechanistic dissection of dynamic cell systems.

## Methods

We describe the COvariate Disentangling Augmented Loss (CODAL) method for the integration of batched single-cell RNA-seq or single-cell ATAC-seq experiments. CODAL specifies a generative model of observed counts which explicitly represents the effects of biological and technical factors in the data. The contributions of these components are disentangled using a novel objective function that regularizes mutual information between the distributions of biological and technical effects. CODAL thereby uncovers an explanation for the influence of technical effects which is maximally independent of biological variation, enabling decomposition of biological effects into interpretable modules, detection of batch-confounded cell states, and representation of cells in a batch-corrected, yet cell type discriminative, latent space.

Because CODAL can detect and represent batch-confounded cell states, this method is useful for the analysis of single-cell perturbation atlases. We infer the parameters of the generative model using fast and memory-efficient streamed gradient ascent over a differentiable approximation of the objective function. The hyperparameters of the model are tuned to a given dataset using highly parallelizable Bayesian optimization. Thus, the method is scalable to large datasets and can leverage more compute resources for faster inference.

In the section "Generative model", we describe the CODAL generative model of observed counts and outline the assumptions made to estimate the rates of expression of genes or chromatin accessibility of genomic loci without technical effect confounding. In "Parameter inference", we define the objective function used to infer the generative model parameters. We then describe differentiable approximations of the terms of the objective function which enables fast inference using stochastic gradient ascent and summarize the complete algorithm used to train a CODAL model. In "CODAL Bayesian hyperparameter optimization", we describe the Bayesian hyperparameter optimization scheme which tunes the model hyperparameters (chiefly the number of topics) to best represent a particular dataset. In "CODAL technical effect augmentation of regulatory potential model", we show how the technical effect estimates of a CODAL model may be used to augment other generative models of observed counts with technical effect correction. "Analysis of NEURIPS bone marrow dataset" details the NEURIPs bone marrow multimodal batch effect benchmark dataset and analyses. In "Frankencell batch-confounded cell type tests", we describe the Frankencell benchmark test for batch-confounded cell types. In "Analysis of mouse embryo differentiation and perturbation", we provide details on the analysis of the mouse embryo differentiation and perturbation dataset. Finally, "Computational resources benchmarking" describes the benchmarking of CODAL's computational resource requirements.

The CODAL generative model is based on the earlier MIRA[28] method for variational autoencoder-based inference of topics from scRNA-seq and scATAC-seq data. CODAL uses the same nonlinear encoder and linear decoder model architectures, latent space reparameterization, and priors as MIRA (sections "Generative model with interpretable biological states and technical effects" and "Lower bound on marginal likelihood"). Unique to CODAL are model components that facilitate technical effect disentanglement, a new parameter inference algorithm and objective function, and a redesigned hyperparameter tuning scheme which is faster and more parallelizable.

### Generative model

The CODAL topic model, an extension of the MIRA topic model, accounts for technical effect confounding biological signal in the analysis of multi-batch scRNA-seq, scATAC-seq, or multimodal data. The model specifies a generative explanation of the cell's observed features (RNA-seq or ATAC-seq reads) as the sum of biological and technical effects. We assume biological effects are purely a reflection of cell state and follow a low-dimensional manifold prescribed by concerted changes in gene regulatory programs in individual cells. Therefore, we model the biological state as a mixture of hidden latent variables which are linearly related to changes in the cell's underlying gene expression or chromatin accessibility state. Each latent variable, or "topic", is associated with a set of linear weights describing the gene expression or genomic loci accessibility changes that are linked to the topic. These topics capture axes of association and covariation evident in the data and suggest the influence of some shared underlying facet of cell state.

**Generative model with interpretable biological states and technical effects.** The technical factors acting on a given batch of cells and their subsequent effects on observed counts are typically unknown, but cell state and technical covariates like batch of origin may serve as proxy variables indicating cells that are influenced by similar factors. We approximate the true distribution of technical effects as a function of these proxies, implemented as a neural network to reflect the unknown functional form of technical effects.

To sample from the generative distribution of transcript counts, $X^{\mathrm{RNA}} \in \mathbb{Z}_{[0,\infty)}^{N_{\mathrm{cells}} \times N_{\mathrm{genes}}}$ (each observation is an element of the set of integers between 0 and infinity) in a multi-batch scRNA-seq dataset with associated cell technical covariates $C \in \mathbb{R}^{N_{\mathrm{cells}} \times N_{\mathrm{covariates}}}$, we first draw from the distribution of the cell state latent random variables $Z \in \mathbb{R}_{[0,1]}^{N_{\mathrm{cells}} \times N_{\mathrm{topics}}}$, where $Z_i$ is a mixture over the latent variables for cell $i$:

$$\sum_{d=1}^{N_{\mathrm{topics}}} Z_{id} = 1, \forall i \in \{1, \ldots, N_{\mathrm{cells}}\} \quad (1)$$

Like Latent Dirichlet Allocation (LDA)[26], we suppose that the cell state mixture of latent variables is Dirichlet-distributed and sparse, such that only a few hidden factors are active in defining the state of each cell. We specify a hierarchical prior that controls the pseudo-counts, $\alpha \in \mathbb{R}_{(0,\infty)}^{N_{\mathrm{topics}}}$, allotted to each latent variable, where $\mathcal{I} \in \mathbb{Z}_{(0,\infty)}$ is

the total pseudocounts allotted to the Dirichlet distribution:

$$Z_{i\cdot} \sim \text{Dirichlet}(\alpha_1, \ldots, \alpha_{N_{\text{topics}}}), \forall i \in \{1, \ldots, N_{\text{cells}}\}$$
$$\alpha_d \sim \text{Gamma}\left(2, \frac{2N_{\text{topics}}}{\mathcal{I}}\right), \forall d \in \{1, \ldots, N_{\text{topics}}\} \quad (2)$$

This allows for data-driven tuning of the prior's sparsity to fit diverse trends and modalities. We fix the total pseudocounts hyperparameter at $\mathcal{I} = 50$ for all $N_{\text{topics}}$ so that the sparsity of the Dirichlet hyperprior is not influenced by the dimensionality of the latent space. Next, we estimate biological effects, or expression rates $\lambda \in \mathbb{R}^{N_{\text{cells}} \times N_{\text{genes}}}$, as a function of cell state $Z$ and topic matrix $\beta \in \mathbb{R}^{N_{\text{topics}} \times N_{\text{genes}}}$, which linearly links the influence of topics to changes in expression space:

$$\lambda_{ij} = \text{batchnorm}(\text{dropout}(Z_{i\cdot}) \times \beta_{\cdot j}),$$
$$\forall i \in \{1, \ldots, N_{\text{cells}}\}, \forall j \in \left\{1, \ldots, N_{\text{genes}}\right\}. \quad (3)$$

The dropout function[52] regularizes the expression model and increases stability during parameter inference. We use the PyTorch[53] implementation of dropout, which sets units in the input matrix to zero at rate $p$, then rescales the matrix by $\frac{1}{1-p}$. The dropout rate parameter is set within a range of $[0.05, 0.1]$ during hyperparameter tuning. The batchnorm function[54] performs the following affine transformation:

$$\text{batchnorm}(Z_{i\cdot}\beta_{\cdot j}) = \gamma_{bj} \frac{Z_{i\cdot}\beta_{\cdot j} - \mu_{Z\beta_j}}{\sigma_{Z\beta_j}} + b_{bj}, \quad (4)$$

which standardizes the product $Z_{i\cdot}\beta_{\cdot j}$ by the mean and standard deviation of the product across all cells for that gene ($\mu_{Z\beta_j}$ and $\sigma_{Z\beta_j}$), then rescales by gene-specific biological effect variance and bias parameters $\gamma_b \in \mathbb{R}^{N_{\text{genes}}}$ and $b_b \in \mathbb{R}^{N_{\text{genes}}}$.

Next, technical effects $t \in \mathbb{R}^{N_{\text{cells}} \times N_{\text{genes}}}$ are estimated as a function of cell state $Z$ and the cell-level covariate proxies $C$. $C$ can include one-hot encoded batch indicator variables and continuous quality control metrics, which could be useful for technical effect disentanglement. The technical effect function is implemented using the neural network $h_\phi$ with weights $\phi$, where the output is zero-centered then rescaled by technical effect variance parameter $\gamma_t \in \mathbb{R}^{N_{\text{genes}}}$ and the technical effect predictions are Bernoulli corrupted with probability 0.05. Bernoulli corruption of technical effects stabilizes cyclical training of the topic model and the mutual information regularizer (see "Parameter inference"). The $h_\phi$ neural network has one input layer with 32 nodes and an output layer. For inputs $Z_{i\cdot}$ and $C_{i\cdot}$ for cell $i$; weights $W_h^0 \in \mathbb{R}^{(N_{\text{topics}} + N_{\text{covariates}}) \times 32}$ and $W_h^1 \in \mathbb{R}^{32 \times N_{\text{genes}}}$; biases $b_h^0 \in \mathbb{R}^{32}$ and $b_h^1 \in \mathbb{R}^{N_{\text{genes}}}$; and layer intermediate $\nu_h^0$, the technical effect for gene $j$ is given by:

$$t_{ij} = \left(h_\phi(Z_{i\cdot}, C_{i\cdot})\right)_j = \begin{cases} \gamma_{tj} \frac{t'_{ij} - \mu_{t'_j}}{\sigma_{t'_j}} & \text{if } c = 0 \\ 0 & \text{otherwise} \end{cases},$$
$$c \sim \text{Bernoulli}\left(\tfrac{1}{20}\right),$$
$$\forall i \in \{1, \ldots, N_{\text{cells}}\}, \forall j \in \left\{1, \ldots, N_{\text{genes}}\right\},$$
$$t'_{i\cdot} = \nu_h^0 W_h^1 + b_h^1,$$
$$\nu_h^0 = \text{dropout} \circ \text{Relu} \circ \text{batchnorm} \circ \left[(\text{dropout}(Z_{i\cdot}) \bigoplus C_{i\cdot})W_h^0 + b_h^0\right],$$
$$\forall i \in \{1, \ldots, N_{\text{cells}}\},$$
$$\phi = \{\gamma_t, W_h^1, b_h^1, W_h^0, b_h^0\}. \quad (5)$$

Above, $\bigoplus$ indicates concatenation and the dropout rate is set to 1/20.

Finally, counts $X^{\text{RNA}}$ from a **scRNA-seq** experiment are drawn from a Negative Binomial noise distribution parameterized by the sum of the biological and technical effects and a gene-level dispersion parameter $\vartheta \in \mathbb{R}^{N_{\text{genes}}}$. Sums of biological and technical effects across all genes in each cell are first transformed into a composition $\rho \in \mathbb{R}^{N_{\text{cells}} \times N_{\text{genes}}}$, where $\sum_{j=1}^{N_{\text{genes}}} \rho_{ij} = 1, \forall i \in \{1, \ldots, N_{\text{cells}}\}$, describing the underlying categorical distribution over transcripts counts in each cell. That composition is scaled by the learned effective read depth, or "size factor", random variable, $n_i$ for cell $i$ to give the rate parameter of the Negative Binomial distribution:

$$X_{ij}^{\text{RNA}} \sim \text{NegativeBinomial}\left(n_i \rho_{ij}, \vartheta_j\right), \forall i \in \{1, \ldots, N_{\text{cells}}\}, \forall j \in \{1, \ldots, N_{\text{genes}}\},$$
$$n_i \sim \text{LogNormal}\left(\log \sum_{j=1}^{N_{\text{genes}}} X_{ij}^{\text{RNA}}, 1\right), \forall i \in \{1, \ldots, N_{\text{cells}}\}, \quad (6)$$
$$\rho_{ij} = \frac{\exp(\lambda_{ij} + t_{ij})}{\sum_{l=0}^{N_{\text{genes}}} \exp(\lambda_{il} + t_{il})}, \forall i \in \{1, \ldots, N_{\text{cells}}\}, \forall j \in \{1, \ldots, N_{\text{genes}}\}.$$

The size factor random variable is given a weak LogNormal prior centered at the observed number of counts measured in that cell. For **scATAC-seq** counts $X^{\text{ATAC}} \in \mathbb{Z}_{[0,1]}^{N_{\text{cells}} \times N_{\text{peaks}}}$, we model observations of accessibility across all regions in each cell using the multinomial distribution:

$$X_{i\cdot}^{\text{ATAC}} \sim \text{Multinomial}\left(\rho_{i\cdot}, \hat{n}_i^{\text{ATAC}}\right), \forall i \in \{1, \ldots, N_{\text{cells}}\}$$
$$\hat{n}_i^{\text{ATAC}} = \sum_{k=1}^{N_{\text{peaks}}} X_{ik}^{\text{ATAC}}, \forall i \in \{1, \ldots, N_{\text{cells}}\}. \quad (7)$$
$$\rho_{ik} = \frac{\exp(\lambda_{ij} + t_{ij})}{\sum_{l=0}^{N_{\text{peaks}}} \exp(\lambda_{il} + t_{il})}, \forall i \in \{1, \ldots, N_{\text{cells}}\}, \forall k \in \{1, \ldots, N_{\text{peaks}}\}$$

Besides the noise distribution, all other aspects of the generative distribution are the same for models of both modalities. In sum, the CODAL topic model is a generative probabilistic latent variable model which describes variation in single-cell count data as the sum of biological and technical effects. The latent topics are modeled as sampled from a sparse Dirichlet-distributed prior and are linearly related to changes in gene expression or peak accessibility. The noise distribution is adapted to fit the specific properties of whichever modality is modeled.

Finally, the unknown distribution of technical effects is approximated using a neural network conditioned on cell state and cell technical covariate proxies. Technical covariates can be provided as either categorical features, which are represented as one-hot vectors, or continuous features, which are standardized. Any number of batches or number of features per batch may be provided.

**CODAL disentangled expression and accessibility predictions.** In using the generative model to estimate gene expression in some cell $i$ without confounding technical effects, we make three assumptions. The first is that technical effects $t$ and expression rates $\lambda$ are mostly independent. We infer a generative model of the data where this is effectively satisfied using our mutual information regularization scheme (described in the section "Lower bound on mutual information"). The second is the expectation over observed technical effects is an unbiased estimator for the true mean effect, and that mean effect has no influence on the gene. Consequently, for many batches measuring cells in the same state, we assume observed counts will be distributed about the true biological measurement of counts.

First, we calculate the expected value of the latent variable composition in some cell, $\bar{Z}_{i\cdot}$, using the posterior distribution:

$$\bar{Z}_{i\cdot} = \mathbb{E}_{z \sim p(\cdot | X_i, C_i)}[Z], \forall i \in \left\{1, \ldots, N_{\text{cells}}\right\}, \quad (8)$$

Then, the unconfounded compositional expression of gene $j$ in that cell, $\bar{\rho}_{ij}$, is given by:

$$\bar{\rho}_{ij} = \frac{\exp(\bar{\lambda}_{ij})}{\sum_{l=0}^{N_{\text{genes}}} \exp(\bar{\lambda}_{il})},$$

$$\bar{\lambda}_{ij} = \gamma_{b,j} \frac{\bar{Z}_i \beta_j - \mu_{Z\beta_j}}{\sigma_{Z\beta_j}} + b_{b,j}, \quad (9)$$

$$\forall i \in \{1, \ldots, N_{\text{cells}}\}, \forall j \in \{1, \ldots, N_{\text{genes}}\}.$$

Recall that we constrain the distribution of sample technical effects to be zero-centered. Thus, $\bar{\lambda}_{ij}$ above is equivalent to the expected expression rate plus the expected value of technical effects:

$$\bar{\lambda}_{ij} = \bar{\lambda}_{ij} + E[t], E[t] = 0. \quad (10)$$

**CODAL latent space and nearest neighbor graph.** The expected value of the posterior distribution of latent topics for cell $i$, $\bar{Z}_{i \cdot}$, falls on the $(N_{\text{topics}} - 1)$ simplex. To analyze distances between cells' state representations on the simplex, we transform the topic compositions to Euclidean vector space using MIRA's[28] isometric log-ratio transformation of $\bar{Z}_{i \cdot}$. We then calculate distances between transformed cell latent variables using the Manhattan distance to create a k-nearest neighbors (k-NN) graph representing cells in similar states. This k-NN graph is used in downstream clustering and UMAP analysis.

## Parameter inference

We aim to learn the distribution of expression rates for cells in state $Z$ without technical effect distortion. From the dependency diagram of the generative model (Fig. 1c), due to the association between $\lambda$ and $t$ implied by their dependence on $Z$, we find $t$ and $\lambda$ are not independent. Consequently, to use our generative model to learn an unconfounded distribution for expression rate, we created a novel objective function, $\mathcal{V}^{\text{CODAL}}$, that penalizes dependence between predicted expression and technical effects, $\lambda$ and $t$, using mutual information[55].

Given a dataset of independent count observations from cells $X^{\text{RNA}}$ or $X^{\text{ATAC}}$ and technical covariates $C$, CODAL learns the parameters of the generative distribution (associations $\beta$, technical effect function parameters $\phi$, dispersions $\vartheta$, and batch normalization parameters $\gamma_b$, $\gamma_t$, and $b_b$) as well as the posterior distribution of cell latent variables $Z$ which maximize:

$$\varphi_{\max} = \text{argmax}_\varphi \sum_{i=1}^{N_{\text{cells}}} \mathcal{V}_\varphi^{\text{CODAL}}(X_i, C_i)$$

$$\mathcal{V}_\varphi^{\text{CODAL}}(X_i, C_i) = \log \int p_\varphi(X_i|Z, C_i) p_\varphi(Z) dZ - I(\lambda, t), \quad (11)$$

$$\varphi = (\beta, \phi, \vartheta, \gamma_b, b_b).$$

This objective finds parameters that both maximize the marginal likelihood of the observations and minimize the mutual information between estimated biological and technical effects over all the cells included in the analysis. However, the optimal parameter values are intractable to compute analytically for both terms, so we approximate these quantities with differentiable lower bounds.

## Lower bound on mutual information

**Mutual information neural estimator.** The CODAL objective function requires that we regularize the mutual information between the distributions of expression and technical effect predictions. Here, we describe a lower bound on mutual information which can be estimated using gradient ascent and neural networks. The mutual information between the distributions, $\mathbb{P}$, of two continuous random variables, in our case $\lambda$ and $t$, is defined by:

$$I(\lambda, t) := D_{\text{KL}}(\mathbb{P}_{\lambda, t} || \mathbb{P}_\lambda \otimes \mathbb{P}_t), \quad (12)$$

Where $\mathbb{P}_{\lambda, t}$ is the joint distribution and $\mathbb{P}_\lambda \otimes \mathbb{P}_t$ is the product of the marginal distributions of the random variables. Mutual information therefore diminishes when the two variables are independent. The Donsker-Varadhan[56] dual form of mutual information is:

$$D_{\text{KL}}(\mathbb{P}_{\lambda, t} || \mathbb{P}_\lambda \otimes \mathbb{P}_t) = \sup_{T \in \mathcal{F}} E_{\mathbb{P}_{\lambda, t}}[T] - \log E_{\mathbb{P}_\lambda \otimes \mathbb{P}_t}[e^T], \quad (13)$$

where the supremum is taken over all functions $T$ in the set $\mathcal{F}$ of all functions for which both expectations are finite. Belghazi et al.[39], approximates the set of functions $\mathcal{F}$ with the set of possible weights, $\Theta$, for a neural network. Thus, they give a lower bound on mutual information called Mutual Information Neural Estimator (MINE), where $T_\theta$ is a neural network with weights $\theta \in \Theta$:

$$D_{\text{KL}}(\mathbb{P}_{\lambda, t} || \mathbb{P}_\lambda \otimes \mathbb{P}_t) \geq \sup_{\theta \in \Theta} E_{\mathbb{P}_{\lambda, t}}[T_\theta] - \log E_{\mathbb{P}_\lambda \otimes \mathbb{P}_t}[e^{T_\theta}]. \quad (14)$$

In practice, for a minibatch of $m$ pairs of $\lambda$ and $t$, corresponding to a sample of $m$ cells, one calculates the lower bound estimate of mutual information, by:

$$I(\lambda, t) \geq \sup_{\theta \in \Theta} \text{MINE}(\lambda, t; \theta, m),$$

$$\text{MINE}(\lambda, t; \theta, m) = \frac{1}{m} \sum_{i=1}^m T_\theta(\lambda_i, t_i) - \log \frac{1}{m^2} \sum_{i=1}^m \sum_{j=1}^m e^{T_\theta(\lambda_i, t_j)}, \quad (15)$$

$$(\lambda_{1 \cdot}, t_{1 \cdot}), \ldots, (\lambda_{m \cdot}, t_{m \cdot}) \sim \mathbb{P}_{\lambda, t},$$

To search the space of $\Theta$, weights $\theta$ at step $s$ are updated by gradient ascent with respect to the mutual information estimate to give a tighter lower bound:

$$\theta^{s+1} \leftarrow \theta^s + \nabla_{\theta^s} \text{MINE}(\lambda, t; \theta^s, m). \quad (16)$$

In summary, the MINE framework trains a neural network to maximize the score difference between paired and unpaired samples of $\lambda$ and $t$, and takes gradient ascent steps with respect to the score difference to tighten the resulting lower bound on mutual information.

We found that using MINE mutual information estimation in the CODAL objective function yielded unstable estimates, likely due to properties of the KL divergence explored by Arjovsky et al.[57] In particular, the KL divergence is not informative for distributions with non-identical support. Therefore, we investigated the Earth Mover, or Wasserstein distance[58] as an alternative measure of the difference between the joint distribution and product of the marginals between two random variables.

**Wasserstein dependency measure.** The Wasserstein dependency measure[38], $I_W$, substitutes the KL divergence in the formulation of mutual information with the Wasserstein distance, $W$, yielding:

$$I_W(\lambda, t) := W(\mathbb{P}_{\lambda, t}, \mathbb{P}_\lambda \otimes \mathbb{P}_t). \quad (17)$$

The Wasserstein distance is defined by the minimum cost of "transporting" the probability density of one distribution onto a second distribution. Typically, this metric is interpreted as the cost of transforming a pile of earth at one location into another at a different location, where the cost incurred is the distance the dirt was moved times the amount. In this scenario, the two piles of dirt are taken to be two distributions, $\mathbb{P}_p$ and $\mathbb{P}_q$, and the Wasserstein distance is the total cost of the most efficient coupling $\gamma$, or transport plan, to convert between them:

$$W(\mathbb{P}_p, \mathbb{P}_q) := \inf_{\gamma \in \Gamma(\mathbb{P}_p, \mathbb{P}_q)} \int_{x \times y} ||x - y|| d\gamma(x, y). \quad (18)$$

Here, $x$ and $y$ are elements of some metric space, the coupling $\gamma$ is a joint distribution of $\mathbb{P}_p$ and $\mathbb{P}_q$, and $\Gamma(\mathbb{P}_p, \mathbb{P}_q)$ is the set of all couplings between the distributions.

Taking $\mathbb{P}_p$ and $\mathbb{P}_q$ to represent our distributions of interest, the Kantorovich-Rubenstein dual form[58] of the Wasserstein distance admits:

$$W\left(\mathbb{P}_{\lambda,t}, \mathbb{P}_\lambda \otimes \mathbb{P}_t\right) = \sup_{T\in\mathcal{L}} E_{\mathbb{P}_{\lambda,t}}[T] - E_{\mathbb{P}_\lambda\otimes\mathbb{P}_t}[T]. \tag{19}$$

where the function $T$ belongs to the set of 1-Lipschitz continuous functions where the expectation is finite: $\mathcal{L}$. We approximate the set of 1-Lipschitz continuous functions using a neural network with weights constrained such that the function defined by the neural network is 1-Lipschitz continuous: $\theta \in \Theta_\mathcal{L}, \Theta_\mathcal{L} = \{\theta|\theta\in\Theta, T_\theta\in\mathcal{L}\}$. In this way, Ozair et al.[38] proposed a lower bound on the Wasserstein dependency measure, $w$, for a minibatch of $m$ pairs of $\lambda$ and $t$:

$$
\begin{aligned}
&I_W(\lambda, t) \geq \sup_{\theta\in\Theta_\mathcal{L}} w(\lambda, t; \theta, m), \\
&w(\lambda, t; \theta, m) = \frac{1}{m}\sum_{i=1}^m T_\theta(\lambda_{i\cdot}, t_{i\cdot}) - \frac{1}{m}\sum_{i=1}^m \log\sum_{i=1}^m \exp T_\theta(\lambda_{i\cdot}, t_{j\cdot}), \\
&(\lambda_{1\cdot}, t_{1\cdot}), \ldots, (\lambda_{m\cdot}, t_{m\cdot}) \sim \mathbb{P}_{\lambda,t}.
\end{aligned}
\tag{20}
$$

These authors found the inclusion of the $\log\sum\exp$ term stabilized training of the estimator. This expression for the lower bound on the Wasserstein dependency metric is a lower bound on MINE plus a constant[40]:

$$w(\lambda, t, \theta, m) \leq \text{MINE}(\lambda, t, \theta, m) + \log m. \tag{21}$$

**CODAL mutual information regularization.** Given the above relationship between mutual information, the MINE estimator, and the Wasserstein dependency measure, we regularize the mutual information between $\lambda$ and $t$ via a lower bound approximation based on the 1-Lipschitz continuous MINE estimator:

$$I(\lambda, t) \geq \sup_{\theta\in\Theta_\mathcal{L}} \text{MINE}(\lambda, t; \theta, m). \tag{22}$$

We use a neural network $T_\theta : \mathbb{R}^{2N_{\text{gene}}} \to \mathbb{R}$, to approximate the function $T$ which satisfies the supremum condition in the dual form of mutation information. We use gradient ascent optimization to determine the network parameters $\theta$ that maximize the function $\text{MINE}(\lambda, t; \theta, m)$. Without 1-Lipschitz constraints on $T_\theta$, the network can maximize this objective by exaggerating small differences between unpaired samples, decoupling the objective score from the apparent strength of the dependence relationship between the variables. This property is remedied by constraining the network to be 1-Lipschitz continuous, as required by Wasserstein distance-based metrics. The 1-Lipschitz constraint also dramatically improves the stability of the gradient with respect to the network parameters, eliminating catastrophic gradient overflows during training.

We employ spectral normalization[59] to enforce 1-Lipschitz continuity of the neural network estimator $T_\theta$, adjusting neural network weights $\theta$ after each gradient step. The $T_\theta$ network, with an input layer, a hidden layer with 64 nodes, and an output layer that outputs a mutual information estimate, is defined as follows:

$$
\begin{aligned}
T_\theta(\lambda, t) &= v_T^1 W_T^2 + b_T^2, \\
v_T^1 &= ReLU\left(v_T^0 W_T^1 + b_T^1\right), \\
v_T^0 &= ReLU\left(\text{dropout}(\lambda\oplus t)W_T^0 + b_T^0\right), \\
\theta &= \left\{b_T^2, b_T^1, b_T^0, W_T^2, W_T^1, W_T^0\right\},
\end{aligned}
\tag{23}
$$

with weights $W_T^0 \in \mathbb{R}^{2N_{\text{gene}}\times 64}$, $W_T^1 \in \mathbb{R}^{64\times 64}$, and $W_T^2 \in \mathbb{R}^{64\times 1}$; and biases $b_T^0 \in \mathbb{R}^{64}$, $b_T^1 \in \mathbb{R}^{64}$, and $b_T^2 \in \mathbb{R}^1$.

**Lower bound on marginal likelihood.** To compute a differentiable lower bound on the marginal likelihood of the data, we use the variational autoencoder[31] approach, which approximates the posterior distribution of $Z$ with the variational distribution $q_\psi(Z|X, C)$. The variational distribution, then, is parameterized by a neural network called the "encoder" with weights $\psi$, which stochastically maps data samples $(X, C)$ to the latent space $Z$:

$$Z \sim q_\psi(\cdot|X, C) = \text{Encoder}_\psi(X, C). \tag{24}$$

This approximation admits the lower bound on marginal likelihood called the "evidence lower bound", or ELBO[31]. For a sample $(X_i, C_i)$ from the dataset:

$$
\begin{aligned}
\log p_\varphi(X_i) &= \log \int p_\varphi(X_i|Z, C_i)p_\varphi(Z)dZ \\
&\geq E_{Z\sim q_\psi(\cdot|X_i,C_i)}\left[\log p_\varphi(X_i|, Z, C_i)\right] - D_{KL}(q_\psi(Z, |X_i, C_i)||p_\varphi(Z)).
\end{aligned}
\tag{25}
$$

We parameterize the variational distribution $q_\psi(Z|X, C)$ such that samples are Dirichlet-distributed using the same method as MIRA. Briefly, for each latent variable in each cell, the encoder neural network outputs mean and variance parameters. These specify a Multivariate Normal distribution with a diagonal covariance matrix. Samples from this Multivariate Normal distribution are transformed into a composition by a Laplace approximation to the Dirichlet distribution[34].

We adapt the architecture of the encoder neural network depending on the modality being modeled. The ***expression encoder neural network*** has an input layer and one hidden layer each comprised of a fully connected feed-forward layer, batch normalization, ReLU activation[60], then dropout. The input and hidden layer have 512 nodes, and the probability of dropout is set to 0.05. Raw expression counts are transformed using deviance residuals[61] before being fed through the encoder model. The output layer is fully connected feed forward followed by batch normalization with mean and variance heads for each latent dimension, plus additional mean and variance heads that parameterize the posterior LogNormal distribution from which the cell-specific size factor parameter, $n_i$, is drawn.

The ***chromatin accessibility encoder neural network*** is implemented using the Deep Averaging Network architecture[62]. Each cell is represented as a "bag of peaks" and encoded as the average of embedding vectors for all peaks in that cell. The average embedding vectors are fed through a hidden layer and an output layer. In addition, a skip connection links the embedding layer to the output layer. Thus, for cell $i$, given embedding matrix $W^0 \in \mathbb{R}^{N_{peaks}\times 256}$; feed-forward weights $W^1 \in \mathbb{R}^{256+N_{\text{covariates}}\times 256}$ and $W^2 \in \mathbb{R}^{256\times 2N_{\text{topics}}}$; and biases $b^1 \in \mathbb{R}^{256}$ and $b^2 \in \mathbb{R}^{2N_{\text{topics}}}$, the accessibility encoder network gives:

$$
\begin{aligned}
\text{Encoder}_\psi(X, C) &= \text{batchnorm}\left((v_{i\cdot}^1 + v_{i\cdot}^0)W^2 + b^2\right) \\
v_{i\cdot}^1 &= \text{dropout}\circ ReLU\circ \text{batchnorm}\circ\left[(v_{i\cdot}^0\oplus C_{i\cdot})W^1 + b^1\right] \\
v_{i\cdot}^0 &= \frac{1}{|\Omega_i|}\sum_{k\in\Omega_i} W_{k\cdot}^0 \\
\Omega_i &= \left\{k|k\in\left\{1, \ldots, N_{peaks}\right\}, X_{ik}^{\text{ATAC}} > 0, \text{Bernoulli}(\tfrac{1}{20}) = 0\right\},
\end{aligned}
\tag{26}
$$

where $\Omega_i$ is the set of accessible peaks in the cell corrupted by leaving out peaks at a rate given by Bernoulli trials with parameter $p = \frac{1}{20}$. The embedding and hidden layer each have 256 nodes.

**Training procedure.** In sections "Lower bound on mutual information" and "Lower bound on marginal likelihood", we specify differentiable lower bounds of the terms of the CODAL objective function. Using those lower bounds, CODAL employs a cyclical stochastic minibatch gradient ascent optimization strategy to learn parameters governing the latent space and generative distribution. First, for the CODAL

objective function:

$$\mathcal{V}_{\varphi}^{\text{CODAL}}(X_i, C_i) = \log \int p_{\varphi}(X_i|Z, C_i)p_{\varphi}(Z)\,dZ - I(\lambda, t), \quad (27)$$

we use the "ELBO" lower bound on the marginal likelihood:

$$
\begin{aligned}
\log p_{\varphi}(X_i) &= \log \int p_{\varphi}(X_i|Z, C_i)p_{\varphi}(Z)dZ \\
&\geq E_{Z \sim q_{\psi}(\cdot|X_i, C_i)}\left[\log p_{\varphi}(X_i|Z, C_i)\right] - D_{KL}(q_{\psi}(Z|X_i, C_i)||p_{\varphi}(Z)),
\end{aligned}
\quad (28)
$$

and the 1-Lipschitz regularized "MINE" lower bound on mutual information between expression rates and technical effects across the whole dataset:

$$I(\lambda, t) \geq \sup_{\theta \in \Theta_{\mathcal{L}}}\text{MINE}(\lambda, t; \theta, m). \quad (29)$$

To estimate mutual information, we draw $m$ samples from the joint distribution of $\lambda$ and $t$ using Monte Carlo sampling from the dataset and the variational distribution of the latent variable Z:

$$
\begin{aligned}
\lambda_{m'} &= Z_{m'}\beta, \quad t_{m'} = h_{\phi}(Z_{m'}, C_i), \\
Z_{m'} &\sim q_{\psi}(\cdot|X_i, C_i), \\
X_{m'}, C_{m'} &\sim \mathbb{P}_{X, C}, \\
\forall m' &\in \{1, \dots, m\}.
\end{aligned}
\quad (30)
$$

Putting together the two lower bounds and introducing term weights $\varepsilon_1$ and $\varepsilon_2$, which are varied over the course of parameter optimization, we define the differentiable approximation of the CODAL objective function $\hat{\mathcal{V}}$:

$$
\begin{aligned}
\hat{\mathcal{V}}(X_i, C_i; \varepsilon_1, \varepsilon_2, m) &= E_{Z \sim q_{\psi}(\cdot|X_i, C_i)}\left[\log p_{\varphi}(X_i|Z, C_i)\right] \\
&- \varepsilon_1 D_{KL}(q_{\psi}(Z|X_i, C_i)||p_{\varphi}(Z)) - \varepsilon_2\text{MINE}(\lambda, t; \theta, m).
\end{aligned}
\quad (31)
$$

Intuitively, the $D_{KL}$ and MINE terms serve to regularize how the model reduces the reconstruction loss term of the ELBO: $E_{Z \sim q_{\psi}(\cdot|X_i, C_i)}[\log p_{\varphi}(X_i|Z, C_i)]$. The $\varepsilon$ coefficients scale the influence of the loss components relative to one another. We anneal $\varepsilon_1$ and $\varepsilon_2$ at every step during training according to cyclically increasing and decreasing schedules, which helps to prevent mode collapse and over-regularization. At step $s$ of training out of a total of training steps $s_{\text{total}}$, $\varepsilon_2$ is set following the cyclic KL annealing schedule[63], $r$, with three cycles:

$$r(s) = \min\left(1, \frac{2 \times \text{mod}(s, s_{\text{total}}/3)}{s_{\text{total}}/3}\right). \quad (32)$$

For $\varepsilon_1$, we implement an annealing schedule we call "step-up cyclic" annealing, $r_{\text{stepup}}$, which linearly increases the maximum value of $\varepsilon_1$ at each cycle:

$$r_{\text{stepup}}(s) = \frac{1}{3}\text{ceil}\left(\frac{s}{s_{\text{total}}/3}\right) \times r(s) \quad (33)$$

Finally, the CODAL objective across the whole dataset may be estimated using subsampled minibatches. Therefore, we use the AdamW[64] optimizer to update the topic model weights by minibatch stochastic gradient ascent. We anneal the learning rate, $\omega_1$, and momentum, $\omega_2$, parameters of the optimizer using the one-cycle learning rate policy during training. The minimum and maximum bounds of the learning rate parameter are determined by the learning rate range test before training. The momentum parameter is annealed between a minimum and maximum of 0.85 and 0.95 during training.

The gradient with respect to the topic model parameters depends on the MINE mutual information estimator. To ensure this model gives reliable estimates of mutual information, we also optimize its

parameters by taking gradient steps with respect to $\theta$ at each step of training using the Adam[65] optimizer (no weight decay) with a learning rate of $10^{-4}$. Parameter updates are followed by spectral normalization to ensure 1-Lipschitz continuity of the estimator.

In sum, we iterate cyclic gradient ascent steps using the following procedure:

> **init** $\varphi, \psi, \theta, \omega_1, \omega_2$
> **for** step $s = 1, \dots, s_{\text{total}}$ **do**:
>
> $\quad \omega_1, \omega_2 \leftarrow$ anneal learning rate and momentum of AdamW optimizer
> $\quad \varepsilon_1, \varepsilon_2 \leftarrow r_{\text{stepup}}(s), r(s)$ (anneal objective term weights)
> $\quad (X^m, C^m) \leftarrow$ randomly sample minibatch of size $m$ from dataset
> $\quad \boldsymbol{g}_{\hat{\mathcal{V}}} \leftarrow \nabla_{\varphi, \psi}\sum_{i=1}^{m}\hat{\mathcal{V}}(X_i^m, C_i^m; \varepsilon_1, \varepsilon_2, m)$ (gradients of topic model weights)
> $\quad \varphi, \psi \leftarrow$ update parameters using $\boldsymbol{g}_{\hat{\mathcal{V}}}$ and AdamW optimizer
> $\quad \boldsymbol{g}_{\text{MINE}} \nabla_{\theta}\text{MINE}(\lambda, t; \theta, m)$ (gradients of MINE estimator)
> $\quad \theta \leftarrow$ update parameters using $\boldsymbol{g}_{\text{MINE}}$ and Adam optimizer
> $\quad \theta \leftarrow \theta/\text{spectralnorm}(\theta)$ (constrain to 1-Lipschitz)
>
> **end for**

By default, we use a minibatch size of 128 and train for a total of 24 iterations over the dataset, or $24 \times \text{ceil}\left(\frac{N_{\text{cells}}}{128}\right)$ total steps. Gradient backpropagation is handled by the PyTorch[53] python package, and we used Pyro's[66] implementation of the ELBO objective. In practice, we estimate mutual information using $m$ samples once per step and use that estimate for gradient calculations for each sample in the minibatch. We use the same mutual information estimate to calculate the gradient with respect to the parameters $\theta$.

## CODAL Bayesian hyperparameter optimization
Single-cell sequencing experiments vary in the number of cells assayed and the biological complexity of the samples. Consequently, a CODAL topic model's fit on a given dataset is dependent on the value for hyperparameter $N_{\text{topics}}$, which sets the dimensionality of the latent space descriptor of biological variation and determines the representational capacity of the model. An appropriate value for $N_{\text{topics}}$ will adequately capture relevant covariance structures in the data and represent all detectable cell types without overfitting. With respect to the number of topics, the CODAL objective score appears to correlate with the analytical quality of the model, meaning hyperparameters that give greater CODAL objective scores also give a better latent representation of the dataset[67]. The $p_{dropout}$ hyperparameter sets the appropriate amount of regularization for producing a good fit for a given dataset. The amount of regularization required correlates with the number of topics and tends to scale inversely with dataset size.

**Bayesian hyperparameter optimization scheme.** Given a dataset $D$, the CODAL objective function $\mathcal{V}^{\text{CODAL}}$, and parameters $\varphi_H$ resulting from training this model with hyperparameters $H$, we aim to find the set of hyperparameters $H_*$ in search space $\mathbb{H}$ which maximizes:

$$H_* = \text{argmax}_{H \in \mathbb{H}}\mathcal{V}_{\varphi_H}^{\text{CODAL}}(D). \quad (34)$$

A set of hyperparameters $H$ is defined as a tuple ($N_{topics} \in \mathbb{Z}_{[1,\infty)}, p_{dropout} \in \mathbb{R}_{[0.05, 0.1]}$) which determines the dimensionality of the latent variable and the dropout noise applied to the linear biological decoder model, respectively. The range of possible values for $N_{topics}$ is set by the user. For example, the optimal number of topics for a typical PBMC dataset ranges from 10 to 20. The NEURIPS bone marrow gene expression dataset contained 22 topics, and the embryo differentiation dataset contained 65 topics.

To find the hyperparameters that give the best fit on a dataset, we use a Bayesian hyperparameter optimization scheme. Sequentially, a set of hyperparameters is sampled, a model is trained with those parameters, and the model is evaluated on a held-out set of cells. Using

the joint distribution of hyperparameters and objective scores from past trials, one predicts scores for untested sets of hyperparameters using a faster-to-compute function. Therefore, one can find and evaluate sets of hyperparameters from subspaces of $\mathbb{H}$ which we expect to yield better models, potentially finding the best set of hyperparameters $H_*$ in fewer steps than by random search.

The CODAL hyperparameter optimization algorithm starts by partitioning the cells in a dataset into training and test portions in a 4:1 ratio, by default, stratified by any covariates supplied by the user. Those partitions are written to disk in fast-loading chunks. For the remainder of the tuning process, data is streamed batch by batch during training to reduce memory overhead. Initially, the tuning algorithm runs 15 startup trails using randomly sampled sets of hyperparameters. In a trial, a CODAL model instantiated with some set of hyperparameters is trained on the training portion of the dataset, and the objective score is evaluated on the testing portion after every epoch. We place checkpoints, or "rungs", after the 8th and 16th epoch, at which the algorithm compares the current trial's objective scores to all previously completed trials that reached that rung. Those in-progress trials which fall in the bottom 50th percentile of scores collected at that rung are "pruned", or their training discontinued. This ensures that computational resources are not wasted on models which are unlikely to offer improvements over trials that have already been computed.

Next, the tuning algorithm switches to Bayesian hyperparameter selection using the algorithm described in the following section ("Bayesian hyperparameter optimization scheme"). We use a Gaussian Process model with a Matern kernel[68] to predict the distribution of scores given hyperparameters. The $\upsilon$ parameter of the Matern kernel is set to 5/2. Tuning iterates for a minimum of 48 trials, after which tuning stops if no improvement in the objective score is recorded in the 12 most recent trials, or if a maximum ceiling of 128 trials is reached. The set of hyperparameters yielding the model with the maximum objective score is taken to best represent the dataset, and that model is returned to the user.

In addition to trial pruning and Bayesian hyperparameter selection, the tuning algorithm employs parallelization of trials to further speed up the tuning process. By default, five concurrent trials may be executed using the SQLite database backend as a message broker, but the optional use of a REDIS database backend enables parallelization to as many cores as desired. We employ the constant liar mean strategy for Bayesian hyperparameter selection during the parallel evaluation of trials. When selecting the next set of hyperparameters to evaluate, there may be many currently running trials for which the objective score is not yet known. The constant liar mean strategy assumes that those trials will result in the mean score of previously completed trials and uses that hypothetical result in the selection of the next set of hyperparameters to evaluate.

This tuning scheme was implemented using the Optuna[69] python package.

**Bayesian optimization acquisition function with trial pruning.** Bayesian optimization is a powerful method for quickly finding sets of hyperparameters that improve model performance on some objective. Briefly, by using a faster-to-compute Bayesian approximation of the objective function, one can sequentially evaluate hyperparameter sets that are expected to perform well based on past trials. At each step, the algorithm chooses the set of hyperparameters which maximizes an "acquisition function" for evaluation in the next trial.

Another popular approach for faster hyperparameter tuning is trial "pruning"[70]. For each set of hyperparameters, a model trains for some number of epochs before the objective score is evaluated. For many problems, the relative performance of models is comparable even before training is complete. Thus, at checkpoints during training

called "rungs", models are scored using the objective function and the worst $q^{th}$ percentage of models are pruned, or their training discontinued. Pruning of poorly performing trials ensures computational resources are not wasted on training models which are unlikely to be competitive.

A tuning algorithm that combines Bayesian optimization with trial pruning could potentially speed up hyperparameter optimization more than either algorithm alone. However, most Bayesian optimization algorithms do not account for the possibility that some trails may not be completed due to pruning. Therefore, we implemented a novel Gaussian Process-based acquisition function that extends the popular "expected improvement" function[71] for use in conjunction with trial pruning.

Given a history of hyperparameter samples $[H_1, \ldots, H_{N_{\text{trials}}}] \in \mathbb{R}^{N_{\text{trials}} \times N_{\text{hyperparameters}}}$ and associated objective scores, $f \in \{\mathbb{R}, \varnothing\}^{N_{\text{trials}} \times N_{\text{rungs}}}$ where each trial has a score for each rung that it reached (and a null score, $\varnothing$, for each that it did not), we select the set of hyperparameters which maximize the $EI_p$, the expected improvement (with pruning), to evaluate in the next trial:

$$H_{N_{\text{trials}}+1} = \text{argmax}_{H \in \mathbb{H}} EI_p(H). \quad (35)$$

Here, "improvement" refers to the improvement of some score $f$ recorded at rung $r$, over the previous best score recorded, $f^*$:

$$\text{Im}(f, r) = \begin{cases} f - f_* & \text{if } f > f_* \text{ and } r = N_{\text{rungs}} \\ 0 & \text{otherwise} \end{cases},$$
$$f_* = \max\{f_{\tau, N_{\text{rungs}}} | \tau \in \{1, \ldots, N_{\text{trials}}\}, f_{i, N_{\text{rungs}}} \neq \varnothing\}. \quad (36)$$

The improvement of a score is zero if it falls below the previous best score or if that score was recorded before the model completed training (reaching the final rung). Expected improvement, then, is the expected value of the improvement function over the joint distribution of scores and rungs for some set of hyperparameters $H$:

$$EI_p(H) = \mathbb{E}[\text{Im}(f, r)] = \sum_{r=1}^{N_{\text{trials}}} \int_{-\infty}^{\infty} \text{Im}(f, r) p_f(f|r, H) df p_r(r|H). \quad (37)$$

The distribution over scores is estimated by the Gaussian Process approximation of the objective. Using the Sklearn Python package's maximum likelihood regressor, we fit a Gaussian Process (GP) distribution on all tuples (score, hyperparameters, rung) from the training history:

$$\left\{ (f_{\tau, r}, H_{\tau}, r) | \tau \in \{1, \ldots, N_{\text{trials}}\}, r \in \{1, \ldots, N_{\text{rungs}}\}, f_{\tau, r} \neq \varnothing \right\}, \quad (38)$$

such that the distribution of scores conditioned on the hyperparameters and the rung is:

$$f|H, r \sim GP(H, r). \quad (39)$$

Because the improvement function, Im, is zero unless the score $f$ is greater than the previous best objective score and was recorded on the last rung, in other words $r = N_{\text{rungs}}$, we can simplify the $EI_p$ function:

$$EI_p(H) = p_r(r = N_{\text{rungs}}|H) \int_{f_*}^{\infty} \text{Im}(f, N_{\text{rungs}}) p_f(f|r = N_{\text{rungs}}, H) df. \quad (40)$$

Next, we reparametrize the distribution for $f$ in terms of the standard normal random variable $z$, where the Gaussian Process function, gp, gives the mean and variance of the score distribution:

$$f|X, r \sim GP(H, r) = N(\mu_{H,r}, \sigma_{H,r}^2) = \sigma_{H,r} z + \mu_{H,r},$$
$$\mu_{H,r}, \sigma_{H,r}^2 = gp(H, r),$$
$$z \sim N(0, 1). \quad (41)$$

We can then transform from $f$-distributed space to $z$-distributed space by standardization:

$$s_{H,r}(f) = \frac{f - \mu_{H,r}}{\sigma_{H,r}}. \tag{42}$$

For $\text{cdf}_z$ and $\text{pdf}_z$ as the cumulative distribution function and probability density function of the standard normal distribution, respectively, the solution to the integral term in the $\text{EI}_p(H)$ formula is given by[71]:

$$\int_{f_*}^{\infty} \text{Im}\left(f, N_{\text{rungs}}\right) p_f\left(f | r = N_{\text{rungs}}, H\right) df = \left(\mu_{H,N_{\text{rungs}}} - f_*\right) \text{cdf}_z(z_*) + \sigma_{H,N_{\text{rungs}}} \text{pdf}_z(z_*). \tag{43}$$
$$z_* = s_{H,N_{\text{rungs}}}(f_*)$$

We contribute the solution to the probability of a trial terminating at a rung given the hyperparameter set $H$. Typically, pruning algorithms work by setting a threshold score that a trial must exceed at each rung, $f_{*r}$. If, at any rung, the threshold is not met, the trial is terminated. The threshold can be the $q^{th}$ percentile of previous scores recorded at rung $r$:

$$f_{*r} = \text{percentile}_q\{f_{i,r} | i \in \{1, \ldots, N_{\text{trials}}\}, f_{i,r} \neq \varnothing\}, \forall r \in \{1, \ldots, N_{\text{rungs}} - 1\}. \tag{44}$$

The probability that a trial terminates at the last rung (or does not terminate at any previous rungs) is given by:

$$
\begin{aligned}
p_r(r = N_{rungs} | H) &= \prod_{r=1}^{N_{\text{rungs}}-1} p_f(f \geq f_{*r} | r, H) \\
&= \prod_{r=1}^{N_{\text{rungs}}-1} 1 - p_z(z \leq s_{H,r}(f_{*r})) \\
&= \prod_{r=1}^{N_{\text{rungs}}-1} \text{cdf}_z(-s_{H,r}(f_{*r})).
\end{aligned} \tag{45}
$$

In summary, given a history of trials and associated scores, we execute the next trial using the set of hyperparameters that maximize the expected improvement acquisition function:

$$\text{EI}_p(H) = \prod_{r=1}^{N_{\text{rungs}}-1} \text{cdf}_z(-s_{H,r}(f_{*r})) \left[\left(\mu_{H,N_{\text{rungs}}} - f_* - \xi f_*\right) \text{cdf}_z(z_*) + \sigma_{H,N_{\text{rungs}}} \text{pdf}_z(z_*)\right], \tag{46}$$
$$z_* = s_{H,N_{\text{rungs}}}(f_*)$$

Above, we add the $\xi$ parameter, which we set at 0.1 by default. This has the effect of overestimating the previous best score when calculating expected improvements, leading to more exploratory hyperparameter choices. For each trial, we test $\text{EI}_p$ for 300 randomly sampled sets of hyperparameters.

## CODAL technical effect augmentation of gene regulation models

Because the technical effects estimated from a CODAL model are effectively independent of the estimates of biological quantities, the technical effect predictions are transferrable: they may be used to augment other scRNA-seq or scATAC-seq read count generative models which do not adjust for technical effects. For example, by fixing the technical effect vectors learned from CODAL as an additive and independent component of the generative distribution for counts, then the biological quantities $\lambda$ may be re-estimated using some other set of features to give $\lambda^{\text{new}}$. The independence assumptions of the generative distribution hold if $\lambda \perp t$ and $\lambda \approx \lambda^{\text{new}}$, as we may reasonably expect $\lambda^{\text{new}} \perp t$.

The subsequent general generative distribution for counts $X_{\cdot j}$ for gene $j$, using noise distribution $\mathcal{D}$, and re-estimated expression rates $\lambda^{\text{new}}$ accounts for technical effect differences between cells. The parameters of this generative distribution may be inferred without

mutual information regularization:

$$
\begin{aligned}
X_{ij} &\sim \mathcal{D}\left(\rho_{ij}^{\text{new}}\right), \\
\rho_{ij}^{\text{new}} &= \frac{\exp(\lambda_{ij}^{\text{new}} + t_{ij})}{\kappa_i}, \\
\lambda_{ij}^{\text{new}} &= f(\ldots), \\
\forall i &\in \{1, \ldots, N_{\text{cells}}\}.
\end{aligned} \tag{47}
$$

Above, $t_{\cdot j}$ is given as a non-trainable vector of technical effects in each cell and the $\kappa_i$ term is a scalar which represents the denominator of the softmax transformation, $\sum_{l=1}^{N_{\text{genes}}} \exp(\lambda_{il}^{\text{new}} + t_{il})$. The CODAL generative model estimates this denominator quantity, which can be used as an approximation for $\kappa_i$:

$$\kappa_i \approx \sum_{l=1}^{N_{\text{genes}}} \exp(\lambda_{il} + t_{il}). \tag{48}$$

In summary, fixing CODAL technical effects as a component of the generative distribution outlined above enables re-estimation of expression rates $\lambda_{\cdot j}^{\text{new}} = f(\ldots)$ while still accounting for differences in technical effects between cells. The parameters of the $f$ function may be inferred without using further mutual information regularization since technical effect confounders have already been disentangled.

## CODAL technical effect augmentation of regulatory potential model

We applied technical effect augmentation to the MIRA regulatory potential (RP) model[28], which relates changes in local chromatin accessibility to gene expression by estimating upstream and downstream exponential decay rates of apparent regulatory influence. The generative model of observed gene expression counts is the same as that used in the CODAL topic model, except expression rates $\lambda_{ij}^{\text{RP}}$ are estimated from the local chromatin accessibility states in cells instead of as a linear function of latent variables. Pure multimodal scRNA-seq and scATAC-seq measurements from the same cells are needed to learn this *cis*-regulatory relationship, so this model is subject to batch effects from both modalities.

Below, $n_i$ is the estimated read depth estimated by the CODAL topic model; $\theta_j^{\text{RP}} \in \mathbb{R}_{(0,\infty)}$ is the dispersion parameter of the Negative Binomial noise distribution; $\mathfrak{D}_{j\eta}$ for $\eta \in \{U, D, P\}$ are the genomic interval sets containing the peaks upstream (U), downstream (D), and proximal to the TSS (the promoter, P); $\delta_i \in \mathbb{Z}_{[0,\infty)}^{N_{\text{genes}} \times N_{\text{peaks}}}$ is the genomic distance in kilobases between every gene and every peak; $a_j \in \mathbb{R}_{(0,\infty)}^3$ are the upstream, downstream, and promoter effect coefficients; $\Delta_j \in \mathbb{R}_{(1,\infty)}^3$ are the upstream, downstream, and promoter decay distances of influence (the promoter has $\Delta_{jP} = \infty$); and $A \in \mathbb{R}_{[0,1)}^{N_{\text{cells}} \times N_{\text{peaks}}}$ is the compositional accessibility rate of peaks in cell. To sample from the RP model generative distribution of observed scRNA-seq counts, $X_{ij}^{\text{RNA}}$, for gene $j$ in cell $i$:

$$
\begin{aligned}
X_{ij}^{\text{RNA}} &\sim \text{NegativeBinomial}\left(n_i \rho_{ij}^{\text{RP}}, \theta_j^{\text{RP}}\right), \\
\rho_{ij}^{\text{RP}} &= \frac{\exp(\lambda_{ij}^{\text{RP}} + t_{ij})}{\kappa_i}, \\
\lambda_{ij}^{\text{RP}} &= \gamma_j^{\text{RP}}\left(\frac{c_{ij} - \mu_{c_j}}{\sigma_{c_j}}\right) + b_j^{\text{RP}}, \\
c_{ij} &= \text{RP}\left(\mathfrak{D}_{j\cdot}, A_{i\cdot}, a_{j\cdot}, \delta_{j\cdot}, \Delta_{j\cdot}\right) = \sum_{\eta \in \{U,D,P\}} a_{j\eta} \sum_{k \in \mathfrak{D}_{j\eta}} A_{ik} 2^{-\delta_{jk}/\Delta_{j\eta}}, \\
A_{ik} &= \bar{\rho}_{ik}^{\text{ATAC}}, \\
\forall i &\in \{1, \ldots, N_{\text{cells}}\}.
\end{aligned} \tag{49}
$$

In extending this model to account for scRNA-seq technical effects, we only add the technical effects $t_j$ for gene $j$ when calculating $\rho_{ij}^{\text{RP}}$. To adjust for scATAC-seq technical effects, we take $A_{ik}$ to equal $\bar{\rho}_{ik}^{\text{ATAC}}$ from the CODAL model, which is the compositional distribution of accessibility without technical effect confounding. The parameters of the technical effect augmented model are estimated using the same MAP inference procedure as MIRA.

**CODAL probabilistic in silico deletion analysis.** We used probabilistic in silico deletion as described in MIRA[28], except we replaced the MIRA regulatory potential generative model with the generative model outline above for tests with technical effect augmentation. The strength of association between a transcription factor motif and a gene is then the difference in likelihood of the generative parameters given all proximal accessible chromatin versus when accessible sites which contain that motif are masked, or in silico deleted. We use a Wilcoxon test to assess the enrichment of association between that motif and set of genes versus background levels of association across all other genes.

## Analysis of NEURIPS bone marrow dataset

**Latent space benchmarking.** We preprocessed gene expression data for the NEURIPS bone marrow[33] using the scanpy[72] python package standard workflow. Batches were concatenated together, cells with fewer than 400 counts and genes with fewer than 30 counts filtered out, cell read depths normalized using the *normalize_total* function with *target_sum* set to 10000, counts log+1 transformed, and highly variable genes found with a minimum dispersion threshold of 0.3, yielding 3500 highly variable genes. Latent spaces for each method were calculated based on the expression of these 3500 genes across all cells. We used sequencing site and donor attributes of cells as technical covariates. We benchmarked CODAL against the MIRA topic model[28], scanpy principal component analysis (PCA), scVI[20], scANVI[18], scanpy PCA+Harmony[19], and scanorama[25]. For scVI and scANVI, we used the same hyperparameters as the scib[21] python package. Harmony and scanorama were tested with default parameters.

For the NEURIPS bone marrow ATAC-seq data, cells with less than 400 peaks and peaks found in less than 30 cells were filtered out. Latent spaces for each method were calculated based on all remaining peaks. We used sequencing site and donor attributes of cells as technical covariates, in addition to the FRiP score for the CODAL model. We benchmarked CODAL against the MIRA topic model, PEAKVI[73], the Scikit-learn[74] implementation of Latent semantic indexing (LSI)[75], and LSI+Harmony. For both GEX and ATAC CODAL topic models, we performed hyperparameter tuning for 32 iterations with the range of possible $N_{\text{topics}}$ set to 15–40.

UMAPs[76] for each latent space were calculated using the UMAP-learn python package, with the *min_dist* parameter set to 0.1 and *negative_sample_rate* set to 3. Silhouette widths[77] for each cell were calculated using the Scikit-learn[74] python package. For cell type label silhouette, we calculated the silhouette width for each cell with respect to the expert-annotated cell type labels provided with the dataset. A higher silhouette width means a cell was more closely grouped with cells of the same label. For batch silhouette width, we calculated silhouette width with respect to the joint label of batch and cell type. In this case, a lower score means a cell is more intermixed with cells from different batches. Average silhouette width was calculated as the mean of silhouette widths across all cells. Both cell type and integration Local Inverse Simpson's Index (cLISI and iLISI)[19] were calculated using the scib package.

**Entangled gene representations.** To assess the influence of disentanglement on expression versus technical effect predictions, we trained a topic model using the same parameters as the CODAL model, but with the weight of the mutual information regularization term of the objective function set to zero.

**Regulatory potential analysis.** We trained regulatory potential models on the multimodal NEURIPS bone marrow dataset for 2841 genes which were both highly variable and had UCSC TSS annotations[78]. We used the same MAP parameter inference procedure as MIRA. To assess the effect of CODAL technical effect augmentation on the likelihood of RP models, we split each of the 13 batches in the dataset into training and test sets in a 4:1 ratio. Then, for each batch, we trained RP models without technical effect augmentation on only the training data of that batch. We also trained RP models with and without augmentation on the combined training set across all batches. Finally, we evaluated the likelihood of the RP models trained under each condition on the test set of each batch.

**Probabilistic in silico deletion analysis.** We selected the top 200 genes from topic 4 to represent a set of genes that were upregulated in Proerythroblast cells in the NEURIPs bone marrow dataset. We evaluated those genes for motif enrichment versus the rest (2641 highly variable genes) using the JASPAR[79] 2020 vertebrate position weight matrix (PWM) collection and probabilistic in silico deletion[28]. For each of 1641 PWMs tested, we compared *p*-values of enrichment assessed with regulatory potential models trained across all batches versus models trained across all batches and augmented with CODAL technical effects. We used Bonferroni-corrected *p*-values to assess the significance of enrichment.

## Frankencell batch-confounded cell type tests

We performed benchmarking to compare CODAL's ability to integrate differentiation trajectories with batch-confounded cell types against popular methodologies. Inspired by the scenario where a wild-type cell atlas suggests future knockout or perturbation experiments, we created a synthetic dataset generation system based on the "Frankencell" python program, where we varied cell abundances along pre-defined trajectory states to introduce varying levels of batch-confounded biology.

**Frankencell benchmark generation.** Frankencell generates synthetic differentiation trajectories by mixing reads from individual cells sampled from distinct, well-defined cell populations from real single-cell RNA-seq or ATAC-seq data. By defining a construction plan in which cell state trajectories interpolate between cell types, Frankencell creates continuous cell state transitions which maintain the statistical properties of counts from real data, but for which the ground truth state of every cell is known. Frankencell can also simulate batch effects by mixing reads from one batch of cells in a multi-batch dataset. In this way, the synthetically mixed cells will also have counts biased by the same technical effects as the reference sample.

For this test, we constructed synthetic datasets using reads sampled from the NEURIPS bone marrow gene expression dataset. We used the expert-annotated cell types as the pure cell type clusters from which reads were mixed. To simulate batch effects, we constructed two trajectories per dataset: one composed of reads from the "site 3, donor 9" batch and one composed of reads from the "site 4, donor 1" batch.

We defined the construction plan as a tree-structured graph, with the root node composed of reads from the HSC cell type and populations that branch to form "lymphoid" and "monocyte" trajectories. The node tree structure and the mixing weights of each node are shown below:

| Nodes | | | Mixing weights | | | | | | |
|---|---|---|---|---|---|---|---|---|---|
| NodeID | Cell type | Parent Node | $\pi_{\text{HSC}}$ | $\pi_{\text{CD16+ Mono}}$ | $\pi_{\text{B1B}}$ | $\pi_{\text{NK}}$ | $\pi_{\text{CD8+T}}$ | $\pi_{\text{pDC}}$ | $\pi_{\text{cDC2}}$ |
| root | "HSC" | | 1 | | | | | | |
| 1 | | root | 0.5 | 0.125 | | 0.125 | 0.125 | | 0.125 |
| 2 | | 1 | 0.2+k | 0.4-k/2 | | | | 0.2-k/4 | 0.2-k/4 |
| 4 | "Mono" | 2 | k | 1-2k | | | | k/2 | k/2 |
| 5 | "Dendritic" | 2 | k | k | | | | 0.5-k | 0.5-k |
| 3 | | 1 | 0.2+k | | | 0.4-k/2 | 0.4-k/2 | | |
| 6 | "B-cell" | 3 | k | | | 1-2k | k/2 | k/2 | |
| 7 | "T-cell" | 3 | k | | | k | 0.5-k | 0.5-k | |

Where mixing weights for a node sum to one (zero weights are left blank), and the k parameter governs the base cell similarity across all cell types. The edges between nodes encode the continuous paths that cells follow through the trajectory structure. To create a smooth transition between states, for a cell that has progressed a fraction $p$ along an edge, its mixing weights were calculated as a sigmoidal transformation, $\sigma$, of the start and end node, $\pi_{\text{start}}$ and $\pi_{\text{end}}$, of that edge:

$$\pi_{\text{cell}} = \pi_{\text{start}}(1-\delta) + \pi_{\text{end}}\delta,$$
$$\delta = \sigma(2p-1). \tag{50}$$

Next, we defined the Markov transition matrix for a cell progressing through the trajectory starting from the root node:

| Node 1 | Node 2 | Transition probability |
|---|---|---|
| Root: "HSC" | 1 | 1. |
| 1 | 2 | 0.4 |
| 1 | 3 | 0.6 |
| 2 | 4: "Mono" | 0.5 |
| 2 | 5: "Dendritic" | 0.5 |
| 3 | 6: "B-cell" | $1 - P_{\text{T-cell}}$ |
| 3 | 7: "T-cell" | $P_{\text{T-cell}}$ |

Where the parameter $P_{\text{T-cell}}$ controls the proportion of cells that transition to the "T-cell" versus "B-cell" terminal states. To generate a synthetic cell, we sampled a path through the trajectory graph starting from the root node and following the Markov transition matrix above until reaching a terminal state. Then, we sampled a progress value according to a beta (0.5,1) distribution to place the cell on an edge of the cell state tree along that path and calculated the cell's read mixing weights based on the sigmoidal interpolation of node-defined mixing proportions. Finally, to sample reads to represent each synthetic cell, we randomly selected one real single cell from each population in the reference dataset and hypergeometrically sampled reads from those cells to fulfill their respective contributions according to the mixing weights. Read depths for each cell were sampled from a LogNormal distribution.

In summary, each Frankencell dataset was composed of two trajectories of 2000 cells, where each trajectory was sampled exclusively from reads from a single batch and generated according to a construction plan with parameters k and $P_{\text{T-cell}}$. In this way, the ground truth cell state and batch identities were known for each cell, and we controlled the difficulty of the test by increasing the base cell-cell similarity and introducing batch-confounded cell types. We generated batch-confounded cell types by first incrementally increasing $P_{\text{T-cell}}$ for batch 1, then fixing $P_{\text{T-cell}}$ at 1 and incrementally decreasing $P_{\text{T-cell}}$ for batch 2. We repeated this depletion process for k ∈ {0, 0.05, 0.1}.

**Performance evaluation on the Frankencell benchmark.** To evaluate the performance of different batch correction methods on these datasets, we calculated latent spaces using counts of highly variable genes and the batch of origin for each cell. We used the MIRA pseudotime trajectory inference algorithm to solve the structure of the resulting latent space and compared it to the ground truth result using established metrics implemented in the dynverse package. Importantly, the trajectory inference algorithm was batch-unaware, so unintegrated batches would score poorly, even if the trajectories within each batch were coherent.

We benchmarked CODAL against PCA+Harmony and scVI. For CODAL, we used standard parameters for tuning, with a topic range of 3-10. We applied the Harmony algorithm to the first 10 principal components calculated using scanpy PCA on log+1 transformed highly variable gene expression counts. For scVI, we used the default parameters used in the scib benchmarking package but found the number of latent dimensions allocated to the model was highly influential on the quality of the latent space. On this dataset, greater latent dimension size resulted in improved marginal likelihoods, even while giving worse solutions to the trajectory. Therefore, we trained models with 3 to 6 latent dimensions and evaluated each for trajectory quality. For each test, we compared the other methods to whichever scVI model scored the highest.

Using the dynverse[48] package, we evaluated integrated trajectories on edge flip, branch F1 score, and pseudotime correlation. Edge flip measures the minimal number of edge additions or subtractions needed to convert the test model's inferred trajectory graph into the ground truth graph, divided by the total number of edges in both graphs (and normalized so that 1 is a perfect score). Pseudotime correlation measures the correlation of temporal geodesic distances between cells in the test versus ground truth trajectories. Branch F1 score quantifies the closeness of the predicted cell state assignment compared to the ground truth. Finally, we calculated iLISI on the Frankencell datasets using the scib package, excluding the "T-cell" and "B-cell" states which are sometimes batch-confounded. The overall score was calculated as the geometric mean of all metrics for each test.

**Evaluation of variance of technical effect estimation.** To analyze the effect of the mutual information regularizer on the repeatability and quality of CODAL representations, we modulated its weight in the CODAL objective by introducing a multiplier weight $\rho$:

$$\mathcal{V}^{\text{CODAL}}_{\varphi,\rho}(X_i, C_i) = \log \int p_\varphi(X_i \mid Z, C_i) p_\varphi(Z) dZ - \rho I(\lambda, t). \tag{51}$$

When this weight is 0, the CODAL objective reduces to marginal likelihood maximization. For each weight $\rho \in \{0, \frac{1}{2}, 1, 2, 4, 8\}$, we trained ten CODAL models with different initial seeds on the same "completely confounded" Frankencell dataset with difficulty k = 0.05. For each model, we collected estimates of biological and technical effects and evaluated the trajectory reconstruction quality using the branch F1 score implemented by dynverse.

Thus, for each weight, we collected data $(\lambda^\rho, t^\rho)$, where $\lambda^\rho$ and $t^\rho$ are tensors of size $(N_{\text{cells}} \times N_{\text{genes}} \times N_{\text{models}})$. To assess the repeatability of CODAL representations, we used variance decomposition across the ten estimates collected for a certain mutual information regularization weight $\rho$:

$$\text{var}\left(\lambda^\rho_{j \cdot}\right) = \mathbb{E}\left[\left(\lambda^\rho_{ij \cdot} - \mathbb{E}\left[\lambda^\rho_{ij \cdot}\right]\right)^2 \mid \text{cell} = i\right] + \text{var}\left(\mathbb{E}\left[\lambda^\rho_{ij \cdot} \mid \text{cell} = i\right]\right), \tag{52}$$

where the first term is the expected squared difference across cells between each estimate and the mean estimate in that gene and that cell across all models, and the second term is the variance across cells of the mean estimate across models. Intuitively, the first term measures how much each model's estimate varies about the average

estimate—the repeatability. We repeated the analysis above for each gene and each $\rho$.

## Analysis of mouse embryo differentiation and perturbation

The mouse embryonic differentiation dataset[8] was preprocessed using the standard scanpy workflow. All batches were concatenated together, cells with fewer than 400 counts and genes with fewer than 30 counts were filtered out, cell read depths were normalized using the *normalize_total* function with *target_sum* set to 10000, counts were log +1 transformed and highly variable genes found with a minimum dispersion threshold of 0.5. CODAL and scanpy PCA latent spaces were calculated from expression of highly variable genes. The CODAL model covariates were defined as the following four categorical variables: the sequencing batch, whether or not the batch was chimeric, whether or not the batch was created from mixing embryos at different points in gastrulation, the day of development at which the batch was collected. We performed 60 iterations of hyperparameter tuning over a number of topics ($N_{topics}$) ranging from 35 to 80. The final model had 63 topics. For the "confounded batch" experiment, we used the same highly variable genes and range of possible $N_{topics}$. UMAPs for each latent space were calculated using the UMAP-learn python package, with the *min_dist* parameter set to 0.1, and *negative_sample_rate* set to 3.

To subcluster chimeric hemato-endothelial progenitor (HE-prog) cells, we performed rough clustering across the whole dataset using the Leiden algorithm and selected a cluster that primarily contained HE-prog cells. Then, we subset again to include only Tom$^+$ *Tal1$^{-/-}$* and ran the Leiden algorithm to obtain high-resolution subclusters. We matched subclusters to mesodermal cell type populations using expression of marker genes provided in Pijuan-Sala et al.[8] and shared topic latent variable composition.

## Computational resources benchmarking

We benchmarked the computational resources required to train CODAL models using publicly-available scRNA-seq and scATAC-seq datasets published by 10x Genomics. For both modes, we simulated datasets of different sizes by downsampling either the number of cells, number of features, or both. For the gene expression model, we trained models on a CPU with each of 4, 8, 16, 32, 64, or 128 topics on datasets with each combination of 1000, 2000, or 4000 features and 2000, 4000, 8000, or 16000 cells. For the chromatin accessibility model, we trained models on an RTX2070 Super GPU with each of 4, 8, 16, 32, 64, and 128 features on datasets with each combination of 50, 100, or 150 thousand features and 1000, 2000, 4000, or 8000 cells. We tracked the total time elapsed and maximum memory used during training.

**Statistics and reproducibility.** No data were excluded from the analyses, and the investigators were not blinded to allocation during experiments and outcome assessment.

## Reporting summary

Further information on research design is available in the Nature Portfolio Reporting Summary linked to this article.

## Data availability

The Frankencell synthetic confounded differentiation datasets generated in this study have been deposited in a Zenodo repository at https://doi.org/10.5281/zenodo.7387471. The NEURIPs bone marrow multimodal dataset is available at https://ftp.ncbi.nlm.nih.gov/geo/series/GSE194nnn/GSE194122/suppl/GSE194122_openproblems_neurips2021_multiome_BMMC_processed.h5ad.gz. Download instructions for the mouse embryo differentiation dataset are available at: https://github.com/MarioniLab/EmbryoTimecourse2018. The single-cell RNA-seq datasets used for computational resources benchmarking are available at https://cf.10xgenomics.com/samples/cell-exp/3.0.0/pbmc_10k_v3/pbmc_10k_v3_filtered_feature_bc_matrix.tar.gz and https://cf.10xgenomics.com/samples/cell-exp/3.0.2/5k_pbmc_v3_nextgem/5k_pbmc_v3_nextgem_filtered_feature_bc_matrix.tar.gz. The single-cell ATAC-seq dataset used for computational resources benchmarking is available at https://cf.10xgenomics.com/samples/cell-exp/3.0.2/5k_pbmc_v3_nextgem/5k_pbmc_v3_nextgem_filtered_feature_bc_matrix.tar.gz. Processed data used in this study are provided in the Source Data file. Source data are provided with this paper.

## Code availability

CODAL is available as part of the MIRA v2.0 Python package at https://github.com/cistrome/MIRA. Models were implemented using Pyro and PyTorch, numerical calculations were implemented using Numpy, and statistical tests were conducted with Scipy[80]. Data is stored in the AnnData structure for interoperability with Scanpy. Package versions were as follows: CODAL core dependencies: pytorch 1.11.0, pyro-api 0.1.2, pyro-ppl 1.8.2. Other packages used in the data analysis presented in the manuscript: anndata 0.8.0, dynwrap 1.2.2. dyneval 0.9.9, dynmethods 1.0.5, matplotlib 3.5.2, numpy 1.21.6, pandas 1.3.4, scanpy 1.9.1, scikit-learn 1.0.2, scipy 1.7.3, seaborn 0.11.2, umap-learn 0.5.3, scib 1.0.2. Packages used in benchmarking: scvi-tools 0.16.4, scvi 0.6.7, harmonypy 0.0.5, scanorama 1.7.2. Synthetic Frankencell datasets were generated using code in this repository: https://github.com/AllenWLynch/frankencell-python. The version of CODAL used for analyses in this study is available at https://github.com/cistrome/MIRA/tree/CODAL, or through: https://doi.org/10.5281/zenodo.7942509[81]. All codes used to produce analyses and figures can be found at https://github.com/AllenWLynch/CODA-reproduction.

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

## Acknowledgements

We thank Prof. Ramesh Shivdasani for the helpful scientific discussions. This work was supported by the National Institutes of Health (NIH) grant U24 CA237617 (C.M.).

## Author contributions

A.L. and C.M. designed the method and analyses. A.L. implemented software and designed figures. A.L. and C.M. wrote the manuscript. C.M. and M.B. supervised the study. All authors read and approved the final manuscript.

## Competing interests

M.B. is a consultant to and receives sponsored research support from Novartis. M.B. serves on the SAB of H3 Biomedicine, Kronos Bio, and GV20 Oncotherapy. The remaining authors declare no competing interests.
