## [Peer Review File · Nature Communications]

Multi-batch Single Cell Comparative Atlas Construction by Deep Learning DisentanglementReviewer #1 (Remarks to the Author):

This work proposes a new computational method called CODAL, which disentangles technical and biological effects for batch integration through a mutual-information-based penalty. CODAL focuses on the integration of cell states across multiple batches, and it allows differences in biological states between batches. Compared with other deep-learning-based approaches developed for related problems, this work has a few significant advantages. First, it achieves a relatively good balance between descriptibility (the network is complex enough to describe complicated datasets) and interpretability. Second, it does not contain too many tuning hyperparameters and gives a clear way to choose them. On several simulated and real datasets, the method shows impressive performance. Overall, the method is solid, the paper is well-written, and enough details are provided.

Major comments:

1. CODAL and a previous method MIRA have very different applications, but they share some ideas in their underlying models (e.g., the topic model). A more detailed introduction/discussion of which parts of the models are shared and which are different may help readers understand CODAL more.

2. Is there an identifiability problem in the CODAL model shown in Figure 1? In other words, are there multiple sets of numerical solutions fitting the data equally well? If there is, how was this problem avoided?

3. On the one hand, "technical effects appear to be jointly dependent on batch and cell state." On the other hand, CODAL "approximates the unconfounded distribution of biological quantities by penalizing the dependence between biological quantities (λ) and technical effects (t)." More discussion/explanation will benefit the readers in understanding this dependence and penalization of dependence. This question is also related to lines 412 to 428 of the Discussion section, where more explanations are also desired---for example, what kind of controls are "appropriate," and are "replicates" required?

4. How demanding is the computation of CODAL? Can the authors give some numbers on the computational time?

Minor comments:

1. More explanation regarding lines 77-80 is needed.

Reviewer #2 (Remarks to the Author):

This paper proposes a new statistical model and computational tool for analyzing multi-batch single cell RNA-seq or ATAC-seq data. The work is motivated by the observation that technical factors in multi-batch single cell data tend to be correlated with true biological signals. Hence a mutual information regularization term was used to augment the usual likelihood to separate biological signals from technical effects. The authors demonstrated that the proposed method can better delineate biological signals compared to existing batch correction methods using simulated benchmarks and real data. The presentation is clear. I have a few comments regarding the statistical model and its assumptions.

One of the key assumptions is that technical effects in any given batch are independent, unbiased samples from the true distribution of technical effects. As the authors mentioned, this may not hold in some settings where the experiments were performed by the same researcher. How sensitive are the model results with respect to this

assumption? Can you still obtain some meaningful results when the assumption is moderately violated?

In the model for the biological signal, the latent variables are Dirichlet-distributed. Could you clarify how sparsity of the latent variables is achieved? Given that the isometric log ratio transformation is used to derive distances in the space spanned by the latent variables, I assume you do not get exact sparsity since you cannot take log ratio of zeros.

What is the pseudocount J (in the prior distribution of α)?

Could you also clarify the definition of the dropout function used in several different places?

In the model for technical effects (t_{ij}), why are the technical effects Bernoulli corrupted?

In the model for X^{RNA} , the total counts per cell was modeled using a log normal distribution with variance 1. Why is the variance fixed at 1?

In the definition of ψ (Section 2 in Methods), the parameter γ_t has already been included in ϕ (parameter for the neural network of the technical effects), unless this γ_t refers to a different quantity.

I understand CODAL works separately for single cell RNA or ATAC-seq data. Can it also jointly analyze the two data types? In this case, the observation per cell is $(X_i^{\text{RNA}}, X_i^{\text{ATAC}})$. I'm curious because the first sentence of the discussion seems to imply such multimodal analysis is feasible with CODAL.

Line 216-217: In general, please be accurate about what the latent variable Z and the weights β refer to. In LDA, the latent variables Z are the topics, while β are the linear association weights. So it is not accurate to call the sets of associations "topics". Similarly, in the first paragraph of Section 1 in Methods, it is better to say that "Each latent variable defines a 'topic', whose weights".

Reviewer #3 (Remarks to the Author):

The authors present CODAL, a variational encoder based model for batch effect identification and batch integration. The key advance is the modification of objective function to include a term that encourages decoupling of batch-specific effects from effects that are conserved across batches that are likely to reflect meaningful biological variation. The method is well-founded mathematically, the manuscript is easy to read, and the software package implementing it seems relatively mature. There are also a series of interesting and diverse benchmark assays addressing some major use cases, and the performance for batch correction appears impressive. Overall the method seems interesting and useful. I have a few minor critiques about usability considerations such as scalability and stability of the approach, and a more conceptual concern about the interpretation of the partitioned batch effects.

1) What happens if one applies CODAL to one batch artificially separated into random splits? Does the model identify no batch effect? If one applies the model to the same dataset multiple times, how well do the batch effect estimates replicate?

2) The software implementation of CODAL appears to be relatively well-engineered. The paper does not include any information about the scalability of the method in terms of time or memory with number of cells, number of batches, etc. Are there any anticipated practical technical limitations?

3) A conceptual question I have is about the relevance of disentanglement as an objective when the batch effect is purposeful – e.g. a drug treatment or change in experimental condition. In this case all cell types are ostensibly affected, and for example my mental partitioning of effects would be:

Expression \sim cell state + direct effect of perturbation + technical effects

where the technical effect should indeed be disentangled from the others, but the direct effect of the perturbation may very well be the most interesting thing to quantify. The demonstrations in Figs. 5 and 6 focus on situations when the batches ultimately are not that different from each other – basically similar up to the spike-in of new or evolving cell states. What would happen if the two batches were +/- a drug treatment? The text includes some hedges about the magnitude of perturbations, but can these be somehow quantified? More generally, the mixing of different types of batch effects into one term clouds interpretation. Perhaps this is not the intended use case for CODAL?

4) More generally, are there any plans to extend the model to incorporate more structured effect models (e.g. as are ubiquitous in traditional differential expression analyses).

Minor comments:

There are some mixups with the figure panel labeling in Figure 1 between the text, the figure, and the legend. Looks like a panel was deleted at some point.

I think a supplemental figure explaining the hyperparameter tuning approach would aid presentation. Similarly I am generally a fan of including tables containing summary statistics for datasets used. E.g. it's not easy to determine the scale of datasets analyzed here. I also had to look through the Github repository to determine how extensible the model was in terms of number and nature of covariates.

Response to Reviews: Multi-batch Single Cell Comparative Atlas Construction by Deep Learning Disentanglement

We thank the reviewers for their careful reading of our manuscript and their constructive criticism. The reviewers' comments are shown in **black**, our responses are in **blue**.

Reviewer #1 (Remarks to the Author):

This work proposes a new computational method called CODAL, which disentangles technical and biological effects for batch integration through a mutual-information-based penalty. CODAL focuses on the integration of cell states across multiple batches, and it allows differences in biological states between batches. Compared with other deep-learning-based approaches developed for related problems, this work has a few significant advantages. First, it achieves a relatively good balance between describability (the network is complex enough to describe complicated datasets) and interpretability. Second, it does not contain too many tuning hyperparameters and gives a clear way to choose them. On several simulated and real datasets, the method shows impressive performance. Overall, the method is solid, the paper is well-written, and enough details are provided.

Major comments:

1. CODAL and a previous method MIRA have very different applications, but they share some ideas in their underlining models (e.g., the topic model). A more detailed introduction/discussion of which parts of the models are shared and which are different may help readers understand CODAL more.

In MIRA we introduced the model architecture for learning topics from single-cell RNA or ATAC-seq data using autoencoding variational inference. Using these topics, MIRA performs various analyses related to the regulation of gene expression. The CODAL method is concentrated on learning topics in batch confounded datasets; all downstream analysis methods implemented in MIRA can be applied to the CODAL results.

The MIRA topic model is composed of a nonlinear neural network encoder, stochastic latent space with a hierarchical Dirichlet prior, and linear decoder mapping from latent back to feature space. A key limitation of the MIRA formulation was that the model did not include adjustment for technical effects for comparing multiple batches. Thus, topics modeled from multiple batches would be contaminated by variance from technical effects. To develop CODAL, we left the core of the MIRA model unchanged but introduced two new components to facilitate batch effect correction: the technical effect decoder network and the Wasserstein dependence regularization network. In addition to these new components, we developed a new parameter inference algorithm using cyclical regularization and momentum annealing and redesigned the hyperparameter selection process to converge faster be more parallelizable and memory efficient.

We have revised the introduction of the manuscript to explicitly state the parts of the model which are shared with the MIRA method.

2. Is there an identifiability problem in the CODAL model shown in Figure 1? In other words, are there multiple sets of numerical solutions fitting the data equally well? If there is, how was this problem avoided?

We thank the reviewer for posing this interesting question about the uniqueness of CODAL solutions when representing batch-confounded cell states. As shown in Figure 1, depending on choices made when constructing a model, the model's representational capacity, and the objective function used, the resulting attribution of variance in a dataset to technical or biological effects can be quite different. A successful objective function should discriminate which models correctly attribute that variance, but since ground truth cell states are unknown, one can only design an objective which assesses whether a model explains the data according to predefined assumptions about the generative process. We show that technical effect modeling by unregularized maximization of the marginal likelihood (ELBO) cannot discriminate between reasonable and unreasonable explanations of the data. Therefore, within the set of models with similar marginal likelihoods, or amongst those that fit the data equally well, there may be many different numerical representations for the generative process behind that data.

One of CODAL's advancements over existing methods is the imposition of structured assumptions about the generative process governing batched single-cell data. These assumptions are designed to ensure that the models which maximize the CODAL objective score account for biological and technical variance accurately and consistently. Because we utilized neural network approximations of the objective function for tractability, CODAL solutions cannot be shown to be globally optimal. Analysis of the conditions under which the CODAL model is identifiable is highly complex, and given the intractability of proving global optimality, it is unclear whether insights from such an analysis would lead to improvements in practice. While we cannot prove identifiability, we empirically demonstrate that unlike ELBO solutions, similarly-scoring CODAL solutions have lower variance with respect to the resulting attributions of biological and technical effects. This follows from the addition of the mutual information regularization, which restricts the model search space to allow only for solutions for which biological and technical effects are minimally dependent on each other.

We also cannot prove the correctness of the resulting set of solutions, as the optimization methods used in CODAL, like other deep learning applications, have no optimality guarantees. Therefore, the training methods used for parameter inference are important; poor training of a good model can result in a poor approximation of the generative process. Consequently, the theoretical performance of the objective function is dependent on the performance of the parameter estimator. We spent considerable effort designing a method for parameter inference using the CODAL objective to maximize the accuracy of the solution. This is challenging to validate using real single-cell data, but we demonstrated using Frankencell simulations that CODAL produces reasonable solutions under challenging conditions.

We address these points with a new experiment outlined in a new figure, Extended Data Fig. 8b,c, where we analyzed the repeatability of CODAL solutions by remodeling the same confounded Frankencell dataset with different initial seedings. We also varied the strength of the mutual information regularization term of the CODAL objective function via a weight to systematically investigate its influence on the generative explanation of the data. We analyzed the variance decomposition of each resulting (cells x genes x models) tensor, conditioned on cells and genes. CODAL's default mutual information regularization strength reduces the expected variance of repeated technical effect estimation 15-fold versus unregularized marginal likelihood maximization. Of the total variance over ten repeated estimations of biological and technical effects, only 0.14% of that variance was attributable to differences between technical effect estimates across models using CODAL, as opposed to 2% for marginal likelihood maximization.

Furthermore, averaged across all models, CODAL attributed 1.5-fold less variance to technical effects compared to marginal likelihood maximization while producing the most faithful confounded trajectory reconstructions. This suggests that the technical effects which were reattributed by CODAL are useful for representing confounded cell states. In sum, the generative assumptions imposed by CODAL, and the design choices made in constructing the model produce repeatable and faithful representations of batch confounded states despite the use of numerical approximation in inferring the parameters of the model.

3. On the one hand, "technical effects appear to be jointly dependent on batch and cell state." On the other hand, CODAL "approximates the unconfounded distribution of biological quantities by penalizing the dependence between biological quantities (λ) and technical effects (t)."
More discussion/explanation will benefit the readers in understanding this dependence and penalization of dependence.

We thank the reviewer for raising this point, which might be confusing to the readers. Our meaning is that technical effects are neither completely independent of cell state, nor are they highly dependent. Without penalizing the dependence, technical effects and cell states will be highly confounded. Therefore, we introduce a mutual information regularization term to mitigate the dependence.

We note that biological quantities are independent of technical effects if all confounding technical factors are observed, and their effects taken into account. In practice, technical factors are unknown, and some factors which correlate with cell state confound direct estimation of biological effects. Our goal is to approximate the unconfounded biological effects by finding a maximally independent representation of technical effects, while allowing some technical effect covariation with expression estimates. Mutual information measures the joint dependence between two distributions, so penalizing this quantity between biological and technical effects enforces a "soft" independence constraint.

Analyzing this regularization scheme in the context of other technical effect correction algorithms demonstrates its effect. A simpler model may find a linear bias term describing the

effect of technical effects on observed data from each batch. The linear bias would satisfy independence between biological and technical effects but would not account for joint cell type and batch dependent technical factors. On the other hand, a model with unconstrained technical effect estimation can capture these effects, but as a side effect, estimates highly correlated and incoherent distributions for biological and technical effects. Our method, using mutual information regularization, balances these extremes based on the structure of the data. The tradeoff between bias and variance in technical effect estimation is chosen by gradient descent optimization of the model parameters with respect to the CODAL objective function. We revised the introduction to clarify that technical effects that vary jointly with biological effects are present in single-cell RNA-seq and ATAC-seq datasets, but they are not the dominant technical signal.

This question is also related to lines 412 to 428 of the Discussion section, where more explanations are also desired---for example, what kind of controls are "appropriate," and are "replicates" required?

To find informative latent spaces, CODAL requires repeated measurements of a subset of cell states that are common to multiple batches. The greater the proportion of cells that share states between batches, the more information there is for technical effect identification. However, what constitutes a sufficient subset of cells in a shared state for effective integration and disentanglement remains an open question. The influence of technical effects on the common cell states informs the model on the variance of technical effect influence on the remainder of the observed data. This helps to disentangle the posterior estimation of biological quantities more robustly.

When analyzing perturbations of cell populations, an appropriate control to include in both treatment and control batches is a known shared population. For example, one should include a group of cells in both batches which are known not to respond to the treatment, or that do not experience the treatment. If a perturbation response varies with cell state or type in the treatment batch, then these signals can be disentangled from cell state independent variation due to technical effects. In addition, inclusion of a known shared cell type allows one to check for under-correction. Ideally, one should also include a positive control: cell states which are known to be different between batches, which functions as a test for overcorrection.

In the discussion section of the revised manuscript, we have expanded on guidelines and limitations for executing perturbation experiments and analyzing the results.

4. How demanding is the computation of CODAL? Can the authors give some numbers on the computational time?

In the revised manuscript we have added a supplementary figure (Extended Data Fig. 1a,b) outlining the computational resources required to train CODAL models. The computational complexity of making an optimal assignment of genes to topics is, in general, NP-hard, but the variational approximation of the posterior of topic assignments as derived in the original Latent

Dirichlet Allocation paper (Blei et al, 2003) has time complexity of roughly $O(NL^2KE)$, where N is the number of samples in the corpus, L is the number of unique features in each sample (for simplicity we assume that each sample has the same number of features), K is the number of topics, and E is the number of iterations through the dataset taken for training.

Empirically, we find similar results for the variational autoencoder-based CODAL model, where, for a fixed number of topics, epochs (which are always fixed at 24), and features, training time-scales approximately linearly with the number of cells in the dataset. With CODAL, however, the training time dependence on the number of features and topics is more complex. Rather than scaling with the number of features represented in each sample, L , our complexity depends on the total vocabulary size V . The training time appears to scale approximately linearly with the number of parameters, P , in the model which is determined by:

$$P(H, V, K, C) = H^2 + (V+2K+C+4)H + (36+C+3K)V + K$$

for encoder hidden layer size H and number of covariates C . Usually, $V \gg H \gg K > C$. Therefore, the training time is more highly influenced by the hidden layer size and the number of features than the number of topics or covariates.

The expected time to train a CODAL model varies with hardware, but as a rule of thumb, a CODAL gene expression model for ~2000 highly variable gene features finishes training in 1-2 minutes per 2000 cells modeled. In practice, a user must train and evaluate 32 or more models using the automatic hyperparameter selection algorithm to find the model configuration which works best for a dataset. Because hyperparameter selection can be parallelized to any number of cores, the total computational time to apply CODAL to a new project is at minimum the time to train any single model, and upper-bounded dependent on the resources that can be utilized (cores and memory).

To reduce memory usage, CODAL caches the dataset to disk at the start of training, then streams batches as needed for stochastic minibatch optimization. Therefore, the memory required to train a model is not dependent on the size of the dataset, only the size of the memory footprint of the model plus a single data "chunk" - preset to contain eight times as many cells as a single minibatch. Typically, we allocate ~1GB of RAM per gene expression model trained in parallel during hyperparameter selection. We have added a section to the introduction which elaborates on the computational considerations for using CODAL and added a method to the package which estimates the time and memory required for model training based on the properties of a dataset.

Minor comments:

1. More explanation regarding lines 77-80 is needed.

We stated that current models do not explicitly model the technical effects influencing samples, and so they cannot prescribe functional forms or regularizations of those technical effects with

respect to the inferred biological variance. Therefore, current methodologies infer mostly unconstrained technical effects. In Figures 1 and 2, we demonstrated unconstrained modeling produces incoherent results where distinct cell states may be erroneously merged and expression estimates are incoherently related to states in which those genes are differentially-expressed. We have expanded the explanation of this limitation in current technologies.

Reviewer #2 (Remarks to the Author):

This paper proposes a new statistical model and computational tool for analyzing multi-batch single cell RNA-seq or ATAC-seq data. The work is motivated by the observation that technical factors in multi-batch single cell data tend to be correlated with true biological signals. Hence a mutual information regularization term was used to augment the usual likelihood to separate biological signals from technical effects. The authors demonstrated that the proposed method can better delineate biological signals compared to existing batch correction methods using simulated benchmarks and real data. The presentation is clear. I have a few comments regarding the statistical model and its assumptions.

1. One of the key assumptions is that technical effects in any given batch are independent, unbiased samples from the true distribution of technical effects. As the authors mentioned, this may not hold in some settings where the experiments were performed by the same researcher. How sensitive are the model results with respect to this assumption? Can you still obtain some meaningful results when the assumption is moderately violated?

The CODAL model decomposes the observed data into biological and technical components:

$$X \sim \exp(\lambda + t)$$

such that, λ and t are minimally dependent, with $E[t] = 0$. Thus, we assume that the observed data from single batches varies about the true distribution of biological states due to variance in technical effects. However, if the technical effects in a group of samples are correlated, then the expected value of those effects will be biased in a way that will, in principle, influence the variation attributed to biological signals.

In practice, biases from correlated technical effects do not seem to irredeemably impair analyses using CODAL. For example, consider a one-batch experiment created under some conditions C_0 from which technical effects are sampled: $t_0 \sim T | C_0$, where T is the distribution of all technical effects. The observed data from this batch is biased, but in our experience researchers will analyze the data and draw biological conclusions that can be validated using orthogonal techniques. In the worst case, a first batch could be influenced by strong effects from the tail of the distribution, which could affect the analysis. If the experiment was repeated in another batch with technical effects sampled under the same conditions: $t_1 \sim T | C_0$, the resulting expected value across all technical effects would be less likely to take extreme values.

Therefore, analyzing more batches, even with correlated technical effects, tends to produce less biased estimates of absolute levels of expression than analysis of a single dataset.

However, even independently collected batches are subject to the underlying and unmeasurable bias of the experimental protocol itself, which suggests that all single-cell experiments generated using the same protocol have some correlated biases. Since correlated biases are so prevalent, we suggest a more practical approach to experimental design should be taken. It is well known that integrating data from diverse sources is more challenging than integrating data generated by the same laboratory using the same batch of reagents. Since the goal of producing a perturbation atlas is ultimately to integrate and compare samples, while minimizing cost, one should pursue the strategy which is most likely to ensure that integration is successful, knowing that the bias of correlated batches is likely to be small compared to the bias of the overall assay. In conclusion, CODAL is not highly sensitive to correlated technical effects, and we do not discourage the extraction of biological inferences from correlated batches. In light of this, we have revised the methods section to reflect these considerations.

2. In the model for the biological signal, the latent variables are Dirichlet-distributed. Could you clarify how sparsity of the latent variables is achieved? Given that the isometric log ratio transformation is used to derive distances in the space spanned by the latent variables, I assume you do not get exact sparsity since you cannot take log ratio of zeros.

The reviewer is correct, in this case we have used the term “sparse” inaccurately. The composition over topics in each cell are modeled as samples from a Dirichlet prior. Samples from this prior are indeed sparse, since they are likely to contain a large fraction of zero values.

During inference, CODAL finds the posterior distribution over latent variables for each cell parameterized as a LogisticNormal distribution. While samples from the posterior LogisticNormal should have a high likelihood under the prior, they do not contain machine-precision zeros. Therefore, CODAL’s posterior estimates of the latent variable compositions are not sparse, but have probability mass concentrated primarily on a few latent variables while most others are allocated very little mass. In practice, this distinction does not affect the interpretability of the model.

We amended the terminology in the manuscript to reflect the more precise definition of sparsity. We thank the reviewer for calling this issue to our attention.

3. What is the pseudocount J (in the prior distribution of α)?

The pseudocount J is the total number of initial pseudocounts distributed across all topics in the symmetric Dirichlet prior. Fixing J for varying numbers of topics enforces similar behavior with respect to the number of topics allocated significant probability mass in each cell for the resulting models. If instead, we set uniform priors with $\alpha=5$ for all models, then models with more topics would produce less concentrated, and therefore less interpretable, posterior distributions.

We clarified the role of this hyperparameter in the methods.

4. Could you also clarify the definition of the dropout function used in several different places?

We use dropout as implemented by the PyTorch python package. At a user-provided rate p (in our case, $p=0.05$) random values of the input matrix are set to zero. Then, the entire matrix is rescaled by a factor of $1/(1-p)$. We added an explanation of the dropout function to the methods document.

5. In the model for technical effects ($t_{\{ij\}}$), why are the technical effects Bernoulli corrupted?

In initial designs of the model, Bernoulli corruption was added to regularize the biological and technical effect estimates such that the biological effects model would produce likely explanations for the observed data even without technical effect correction. Later, the injection of noise into the technical effect estimates was found to be important for the stability of CODAL during training.

Training the CODAL model proceeds with cyclical updates to the topic model weights and the dependence model weights, where the two models are engaged in an adversarial game. The topic model update seeks to reduce the apparent mutual information between biological and technical effect estimates according to the current state of the dependence model. Then, the dependence model is updated to tighten the lower bound with respect to the true mutual information between biological and technical effects. We found using uncorrupted, or noiseless, empirical samples from biological and technical effects to update the dependence model weights caused the dependence model to learn too fast and to become too strong a regularizer. This often led to mode collapse during training, as the topic model weights were driven into poor local minima shaped by severe penalties from the dependence network.

Adversarial training requires that both the topic model and dependence model explore objective space for a jointly-amenable solution - one in which the topic model expressively represents biological variance without violating the conditions of the mutual information regularizer. Noise injection allows for simultaneous training of these models which, empirically, guides them to jointly-amenable minima.

We thank the reviewer for raising this important question. We have elaborated on the role of the Bernoulli corruption in the methods.

6. In the model for $X^{\{RNA\}}$, the total counts per cell was modeled using a log normal distribution with variance 1. Why is the variance fixed at 1?

The total counts per cell in scRNA-seq data are modeled as a free “size factor” parameter estimated for each cell. Its posterior is given by mean and variance heads of the encoder neural network. The size factor was given a weak prior specified by the lognormal distribution with mean centered at the log-total counts in each cell and variance 1. The variance could be shrunk empirically, but this had no discernible effect on model quality. We changed the methods document to make clear that the size factor parameter is estimated by the gene expression encoder network, and that the aforementioned distribution functions as a weak prior. We thank the reviewer for calling this incomplete explanation to our attention.

7. In the definition of ψ (Section 2 in Methods), the parameter γ_t has already been included in ϕ (parameter for the neural network of the technical effects), unless this γ_t refers to a different quantity.

γ_t does not refer to a new parameter. We thank the reviewer for finding this error. We fixed the notation in this section to clarify this.

8. I understand CODAL works separately for single cell RNA or ATAC-seq data. Can it also jointly analyze the two data types? In this case, the observation per cell is $(X_i^{\text{RNA}}, X_i^{\text{ATAC}})$. I'm curious because the first sentence of the discussion seems to imply such multimodal analysis is feasible with CODAL.

This was confusing as originally written, as we do not intend that the method be used this way. We have designed models which work independently for both scRNA-seq and scATAC-seq data and previously demonstrated how these data views can be merged to construct a joint representation of multimodal data. We pursued a strategy of merging independent analyses of each mode to identify states in which gene expression and chromatin accessibility may be changing in different or unexpected ways. As we and others have found, the timing of cell state changes along a differentiation trajectory can appear different as measured by each modality. We have clarified this in the introduction of the manuscript.

9. Line 216-217: In general, please be accurate about what the latent variable Z and the weights β refer to. In LDA, the latent variables Z are the topics, while β are the linear association weights. So it is not accurate to call the sets of associations “topics”. Similarly, in the first paragraph of Section 1 in Methods, it is better to say that “Each latent variable defines a ‘topic’, whose weights”.

We thank the reviewer for calling this to our attention. We amended the manuscript so that the latent variables are referred to as topics and the linear association weights are called “associations”.

Reviewer #3 (Remarks to the Author):

The authors present CODAL, a variational encoder based model for batch effect identification and batch integration. The key advance is the modification of objective function to include a term that encourages decoupling of batch-specific effects from effects that are conserved across batches that are likely to reflect meaningful biological variation. The method is well-founded mathematically, the manuscript is easy to read, and the software package implementing it seems relatively mature. There are also a series of interesting and diverse benchmark assays addressing some major use cases, and the performance for batch correction appears impressive. Overall the method seems interesting and useful. I have a few minor critiques about usability considerations such as scalability and stability of the approach, and a more conceptual concern about the interpretation of the partitioned batch effects.

1) What happens if one applies CODAL to one batch artificially separated into random splits? Does the model identify no batch effect? If one applies the model to the same dataset multiple times, how well do the batch effect estimates replicate?

We tested the first scenario on a publicly available single batch dataset of five thousand PBMC cells produced by 10x Genomics. We created clusters using PCA followed by the Leiden clustering algorithm to find structured communities of cells. We then randomly divided the dataset into two false batches and modeled the data using CODAL. Firstly, CODAL found a latent space in which the false batches were well mixed (Fig. R1. top left), but also in which the overall cell state structure of the dataset was well preserved (Fig. R1 top right). For T-cell, NK cell, and dendritic cell markers CCL5, GNLY, and CST3, respectively, CODAL inferred that technical effects were not different between false batches (Fig. R1, bottom).

To address the second scenario, we conducted a new experiment outlined in Extended Data Fig. 8 b,c. We thoroughly address the identifiability and repeatability of CODAL solutions in the response to reviewer 1, comment 2, and briefly

describe the results of the experiment here:

We analyzed the repeatability of CODAL solutions by remodeling the same confounded Frankencell dataset with different initial seedings. We also varied the strength of the mutual

information regularization term of the CODAL objective function via a weight to systematically investigate its influence on the generative explanation of the data. Using variance decomposition of the each resulting (cells x genes x models) tensor, conditioned on a cell and gene, we found that CODAL's default mutual information regularization strength reduces the expected variance of repeated technical effect estimation 15-fold versus unregularized marginal likelihood maximization. In fact, of the total variance describing ten repeated estimations of biological and technical effects, only 0.14% of that variance was attributable to differences between technical effect estimates across models using CODAL, as opposed to 2% for marginal likelihood maximization.

Furthermore, averaged across all models, CODAL attributed 1.5-fold less variance to technical effects compared to marginal likelihood maximization while producing the most faithful confounded trajectory reconstructions, which suggests those technical effects which were reattributed by CODAL are useful for representing confounded cell states. In sum, the generative assumptions imposed by CODAL and the design choices made in constructing the model produce repeatable and faithful representations of batch confounded states despite the use of numerical approximation in inferring the parameters of the model .

We thank the reviewer for suggesting these informative ways to interrogate the performance of the CODAL model.

2) The software implementation of CODAL appears to be relatively well-engineered. The paper does not include any information about the scalability of the method in terms of time or memory with number of cells, number of batches, etc. Are there any anticipated practical technical limitations?

We refer the reviewer to reviewer 1, comment 4, where we address the time complexity and practical aspects of using the CODAL model. Briefly, CODAL training time is approximately linearly complex with respect to the number of cells in a dataset and the number of parameters of the model. Model size is mostly determined by the dimensionality of the feature space (highly variable genes for expression or called peaks for chromatin accessibility datasets) and the hidden layer size of the encoder network. Memory requirements are constant with respect to the number of cells because CODAL trains from an on-disk cache where cells are loaded into memory as needed for streaming minibatch stochastic gradient descent. Finally, the hyperparameter tuning algorithm is parallelizable to any number of cores. CODAL requires about 1-2 minutes of training per 2000 cells to model a typical scRNA-seq dataset with 2000 highly variable genes.

3) A conceptual question I have is about the relevance of disentanglement as an objective when the batch effect is purposeful – e.g. a drug treatment or change in experimental condition. In this case all cell types are ostensibly affected, and for example my mental partitioning of effects would be:

Expression ~ cell state + direct effect of perturbation + technical effects

where the technical effect should indeed be disentangled from the others, but the direct effect of the perturbation may very well be the most interesting thing to quantify. The demonstrations in Figs. 5 and 6 focus on situations when the batches ultimately are not that different from each other – basically similar up to the spike-in of new or evolving cell states. What would happen if the two batches were +/- a drug treatment?

When the technical effects and the biological effects are completely confounded an absolute correction of technical effects would be required to determine the biological effect of the perturbation. This goes beyond the application we are proposing for CODAL, which does not model the underlying technical factors in an absolute sense.

If the two batches involve a drug treatment and a control, the capacity of CODAL to correct the technical effects would depend on the extent to which the drug impacts different cell states. It is likely that most drugs at clinically relevant doses would have a larger impact on some cell states than others, and that a sufficient proportion of cell states would be shared between batches to allow CODAL to disentangle technical and biological effects. Stronger drug treatments might shift all cell states to a degree that no longer allows for a meaningful comparison.

CODAL requires repeated measurements of a subset of cell states that are common to multiple batches. For an experiment to satisfy this condition, one can include a group of cells which does not experience or do not respond to a perturbation in both control and treatment batches. Other strategies to create batches with cell-state dependent perturbation include multiplexing treatment samples with control groups, or using genetic perturbations which affect a subset of cells. In summary, with simple experimental considerations, the disentanglement objective is empowered to discover perturbation-induced cell states.

The reviewer notes that the batched datasets we analyzed shared such a large proportion of similar states between batches that the confounded cell states constituted a spike-in. Generally, a spike-in refers to the incorporation of a minute amount of substance possessing innate features that facilitate its differentiation from the rest of the material. Our tests are akin to a spike-in with respect to incorporating a limited number of cells, but they significantly differ from a spike-in concerning their resemblance to the remaining cells. We intentionally designed the tests in this manuscript to satisfy the conditions for inference outlined above, but also to challenge the cell state detection sensitivity of CODAL and existing methods. Discriminating between highly similar but non-identical population distributions is more challenging than discrimination between dissimilar or disjoint cell states. While existing methods perform satisfactorily when comparing highly dissimilar groups of cells between batches, we showed there exists a need for a more sensitive tool when comparing batches in the scenario that perturbations cause more subtle differences.

We did compare datasets with highly dissimilar cell types, however, in the “confounded batch” embryonic differentiation dataset test (Fig. 6, Extended Data Fig. 9a,b), where we removed overlapping cell populations between batches. We demonstrated that CODAL is robust to highly

different (and even disjoint) population distributions between batches, and can still discover cross-batch temporal relationships between cell states in this challenging situation.

We thank the reviewer for posing these constructive questions which demonstrate when CODAL may be used. We expanded the discussion section to detail how batches may be constructed such that CODAL can disentangle effects of perturbations.

The text includes some hedges about the magnitude of perturbations, but can these be somehow quantified? More generally, the mixing of different types of batch effects into one term clouds interpretation. Perhaps this is not the intended use case for CODAL?

Assessing the absolute magnitude of a perturbation in gene expression space from batched data is difficult due to the presence of confounding technical effects. The CODAL latent space, however, represents the inferred biological variance partitioned from technical effects in a group of batched samples. CODAL latent variables relate coherently to changes in gene expression/chromatin accessibility space via the matrix of linear associations β , and so inter-cell distances in the latent space may be a useful measure of the magnitude of perturbation across batches. Comparing and measuring magnitudes of perturbations is out of the scope of this study, but presents an interesting potential application for CODAL.

The reviewer raises an interesting possibility of characterizing different contributions of batch effects. In the current CODAL model we treat the technical effect model as a black box. Reconfiguration of the technical effect model to facilitate the interpretation of these effects is a subject we will pursue in future work.

4) More generally, are there any plans to extend the model to incorporate more structured effect models (e.g. as are ubiquitous in traditional differential expression analyses).

Our goal has been to disentangle biological and technical effects to facilitate a broader range of downstream analyses on batched single-cell RNA-seq and ATAC-seq data. We paid special attention to ensure our model worked well for structured and hierarchical batch effects. We hope that the biological representations found by CODAL will be useful for post-hoc structured analysis of effects of perturbations, differential abundance, etc. We see the statistical analysis of the biological relationships as separate from the technical effect disentanglement analysis, and currently do not plan to build more specialized structured CODAL models.

Minor comments:

There are some mixups with the figure panel labeling in Figure 1 between the text, the figure, and the legend. Looks like a panel was deleted at some point.

We have corrected the discrepancies in the manuscript. We thank the reviewer for finding these errors.

I think a supplemental figure explaining the hyperparameter tuning approach would aid presentation. Similarly I am generally a fan of including tables containing summary statistics for datasets used. E.g. it's not easy to determine the scale of datasets analyzed here. I also had to look through the Github repository to determine how extensible the model was in terms of number and nature of covariates.

Please refer to the figure which we added to the supplement (Extended Data Fig. 1c) to explain the hyperparameter tuning approach. We also added the supplementary table below to the manuscript which describes the structure and size of the datasets used. Finally, the CODAL model can take any number of categorical or continuous features as covariates. Categorical covariates, like batch of origin, are converted to one hot vectors. Continuous features are standardized. We added this information to the methods.

Dataset	Data type	Number of cells	Number of batches	Covariates modeled
NEURIPS bone marrow	Pure multimodal scRNA-seq + scATAC-seq	60,000	13	Donor, sequencing center, FRIP score (ATAC only)
Frankencell	Simulated scRNA-seq	4,000	2	Batch only
Embryonic differentiation	scRNA-seq	150,000	6	Batch, "Is wildtype", "Is chimera", "Is mixed embryos" - one batch was created by mixing together cells from multiple embryos.

Reviewer #1 (Remarks to the Author):

The authors have successfully addressed all my previous concerns.

Reviewer #2 (Remarks to the Author):

The authors have addressed my previous concerns satisfactorily.

Reviewer #3 (Remarks to the Author):

I have no major follow-up comments. The paper was in fairly good shape when it was submitted and I am satisfied with the response to reviewer comments. I recommend accepting the paper.

Response to Reviews: Multi-batch Single Cell Comparative Atlas Construction by Deep Learning Disentanglement

We thank all the reviewers for their careful reading of our manuscript and their constructive criticism. The reviewers' comments are shown in **black**, our responses are in **blue**.

REVIEWERS' COMMENTS

Reviewer #1 (Remarks to the Author):

The authors have successfully addressed all my previous concerns.

We are glad that we were able to address all your concerns satisfactorily. Your feedback has helped us improve the quality and clarity of our paper.

Reviewer #2 (Remarks to the Author):

The authors have addressed my previous concerns satisfactorily.

We are pleased to hear that we have addressed your previous concerns to your satisfaction. Your feedback has been invaluable in refining our work, and we are grateful for your input.

Reviewer #3 (Remarks to the Author):

I have no major follow-up comments. The paper was in fairly good shape when it was submitted and I am satisfied with the response to reviewer comments. I recommend accepting the paper.

We are pleased to know that you found our paper in fairly good shape when it was submitted, and that you are satisfied with our response to the reviewer comments. We appreciate your recommendation to accept the paper.